# Rethink the Role of Neural Decoders in Quantum Error Correction

Ge Yan [1]   Shanchuan Li [2 1]   Yuxuan Du [1 3]

## Abstract

Quantum error correction (QEC) is essential for enabling quantum advantages, with decoding as a central algorithmic primitive. Owing to its importance and intrinsic difficulty, substantial effort has been made to QEC decoder design, among which neural decoders have recently emerged as a promising data-driven paradigm. Despite this progress, practical deployment remains hindered by a fundamental accuracy–latency tradeoff, often on the microsecond timescale. To address this challenge, here we revisit neural decoders for surface-code decoding under explicit accuracy–latency constraints, considering code distances up to $d = 9$ (161 physical qubits). We unify and redesign representative neural decoders into five architectural paradigms and develop an end-to-end compression pipeline to evaluate their deployability and performance on FPGA hardware. Through systematic experiments, we reveal several previously underexplored insights: (i) near-term decoding performance is driven more by data scale than architectural complexity; (ii) appropriate inductive bias is essential for achieving high decoding accuracy; and (iii) INT4 quantization is a prerequisite for meeting microsecond-scale latency requirements on FPGAs. Together, these findings provide concrete guidance toward scalable and real-time neural QEC decoding.

## 1. Introduction

Recent experimental milestones of quantum error correction (QEC) have demonstrated *surface codes* (Fowler et al., 2012) on superconducting quantum processors, indicating that logical errors can be continuously suppressed through reduced physical error rates, increased code distance, and increasingly effective decoding (Bausch et al., 2024; Gao et al., 2025). However, extending these proof-of-concept demonstrations to scalable and practical fault-tolerant quantum computing is highly challenging, as theoretical results establish that optimal decoding is NP-hard in general (Iyer & Poulin, 2015). Consequently, practical QEC decoding necessarily operates in a near-optimal regime, governed by *a critical tension between decoding accuracy and real-time efficiency*. In particular, to sustain logical qubits beyond the coherence limits of superconducting circuits, surface-code decoding must deliver sufficiently high accuracy to enable exponential error suppression while simultaneously meeting stringent microsecond-scale latency constraints (Terhal, 2015; Das et al., 2022; Barber et al., 2025).

The central role of QEC decoding has motivated extensive efforts to design effective decoders for surface codes. In this context, traditional heuristic decoders such as minimum-weight perfect matching (MWPM) (Dennis et al., 2002) and belief propagation (BP) (Poulin & Chung, 2008) capture specific points in this tradeoff, but struggle in general and real-time settings (Terhal, 2015). Recently, learning-based decoders have emerged as a promising alternative (Alexeev et al., 2025), offering data-driven mechanisms to navigate these tradeoffs under realistic noise and system constraints. In this context, a variety of neural decoders have been proposed, exploring diverse neural architectures and input representations of error information.

Despite this proliferation, how neural decoding should resolve the fundamental tension between decoding accuracy and real-time efficiency remains largely elusive. In particular, many neural decoders report improved performance over classical baselines. This raises the first key question.

Q1: *Do these gains primarily come from architectural design, or are they driven by increased training data?*

Crucially, the answer to Q1 has direct implications for real-time feasibility of neural decoders. That is, most existing neural decoders are developed to pursue high accuracy, and consequently overlook real-time efficiency. While some studies advocate mitigating this limitation through deployment on specialized hardware (e.g., FPGAs), or by adopting neural compression techniques (Gholami et al., 2021),

[1]College of Computing and Data Science, Nanyang Technological University, Singapore 639798, Singapore [2]Department of Electrical Engineering and Computer Science, Tokyo University of Agriculture & Technology, Koganei, Tokyo, 184-8588, Japan [3]School of Physical and Mathematical Sciences, Nanyang Technological University, Singapore 639798, Singapore. Correspondence to: Yuxuan Du <duyuxuan123@gmail.com>.

*Proceedings of the 43rd International Conference on Machine Learning*, Seoul, South Korea. PMLR 306, 2026. Copyright 2026 by the author(s).

their practical feasibility and deployment readiness remain largely unverified. These limitations naturally motivate a second key question.

Q2: *How can neural decoding be made compatible with stringent real-time efficiency requirements in practice?*

In this work, we reconcile the tension between decoding accuracy and real-time efficiency by revisiting neural decoder design through the lens of Q1 and Q2. For Q1, we reproduce and rigorously evaluate five representative neural decoder architectures, i.e., multilayer perceptrons (MLP), convolutional neural networks (CNN) (Varsamopoulos et al., 2017), temporal convolutional networks (TCN) (Chamberland & Ronagh, 2018), Transformers (Senior et al., 2025), and graph neural networks (GNN) (Liu & Poulin, 2019). The experiments are conducted on surface codes with distances up to $d = 9$, corresponding to systems with up to 161 physical qubits, which lie at the frontier of current experimental capabilities and are relevant for near-term fault-tolerant demonstrations. For Q2, we develop a complete model compression pipeline that integrates weight pruning with neural quantization (Gholami et al., 2021), including both post-training quantization (PTQ) (Nagel et al., 2020) and quantization-aware training (QAT) (Jacob et al., 2018), enabling a systematic evaluation of real-time efficiency without sacrificing decoding accuracy. Building on the resulting lightweight neural decoders, we further analyze their feasibility for deployment on FPGA platforms under realistic resource and latency constraints.

Our experiments, based primarily on large-scale Stim simulations (Gidney, 2021) and validated through fine-tuning on publicly available Sycamore data (AI, 2023) at distances $d = 3, 5$, yield several pivotal insights relevant to Q1 and Q2. For Q1, experimental results indicate that decoding accuracy gains in the near-term are driven disproportionately by dataset scale rather than architectural complexity. In particular, a simple neural decoder trained on a large-scale dataset of $10^7$ samples consistently outperforms more complex architectures trained on standard-sized datasets. These findings suggest that **QEC decoding operates in a data-first regime**, which has not been systematically characterized in prior work. As a side result, experimental results uncover that inductive bias is non-negotiable in neural decoding. Concretely, generic MLPs fail to scale, while GNN-based neural belief propagation struggles with the short-cycle structure of surface codes. In contrast, decoders that combine local convolution with sequential aggregation consistently provide the most robust performance.

For Q2, our results reveal an **efficiency sweet spot** for near-term fault tolerance ($d \leq 9$), in which lightweight architectures with $\sim 10^5$ parameters suffice to achieve near-saturated decoding fidelity for surface codes in this regime. Moreover, we show that aggressive quantization to **INT4 is**

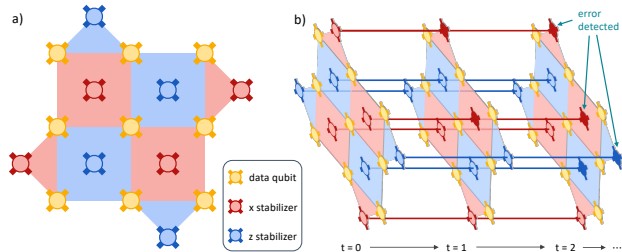

*Figure 1.* **Schematic of a** $d = 3$ **rotated surface code.** a) Data qubits (yellow) are located at the vertices, interleaving with two types of detector qubits (stabilizers) located on the faces. Red squares represent x-stabilizers (detecting z errors), and blue squares represent z-stabilizers (detecting x errors). b) As time goes on, the errors accumulate on the detectors.

**essential for meeting real-time latency constraints**, rather than serving as a post hoc optimization. Such quantization is **only achievable through QAT**, underscoring that hardware constraints must be explicitly incorporated into the training process to enable real-time decoding. Taken together, these results demonstrate that accuracy and latency must be co-designed in neural QEC decoding, with hardware constraints explicitly informing model design and training.

## 2. Preliminary

Here we first introduce the QEC and the rotated surface code, then explain how to reformulate QEC decoding as a binary classification task. Refer to Appendix A for more details on real-time decoding and FPGA.

**QEC and stabilizer codes**. Quantum error correction (QEC) protects quantum information by encoding $k$ logical qubits into $n$ physical qubits and is essential for achieving quantum advantage (Gottesman, 2009). A QEC code is commonly specified by parameters $[\![n, k, d]\!]$, where the code distance $d$ determines the number of errors that can be reliably detected and corrected.

Among the various QEC codes developed over the past decades, stabilizer codes are particularly prominent, owing to their well-defined mathematical structure and effective implementations. The central idea is to characterize an $n$-qubit quantum state $|\psi\rangle \in \mathbb{C}^{2^n}$ by a set of Pauli operators of which the state is invariant. Let $\mathcal{P}_n = \{\pm 1, \pm i\} \times \{I, X, Y, Z\}^{\otimes n}$ be the $n$-qubit Pauli group with $I = \left(\begin{smallmatrix} 1 & 0 \\ 0 & 1 \end{smallmatrix}\right)$, $X = \left(\begin{smallmatrix} 0 & 1 \\ 1 & 0 \end{smallmatrix}\right)$, $Y = \left(\begin{smallmatrix} 0 & -i \\ i & 0 \end{smallmatrix}\right)$, $Z = \left(\begin{smallmatrix} 1 & 0 \\ 0 & -1 \end{smallmatrix}\right)$. A stabilizer group $\mathcal{S} \subset \mathcal{P}_n \setminus \{-I\}$ is an Abelian subgroup, and $|\psi\rangle$ is stabilized by $\mathcal{S}$ if $S|\psi\rangle = |\psi\rangle, \forall S \in \mathcal{S}$. The stabilizer group $\mathcal{S}$ provides a natural framework for QEC, with $|\mathcal{S}| = n - k$. A physical error that *anticommutes* with at least one stabilizer $S \in \mathcal{S}$ flips the corresponding stabilizer measurement outcome, producing a nontrivial **syndrome** $s \in \{0, 1\}^{n-k}$. By measuring the stabilizers and extracting this syndrome, one can infer an appropriate correction operation and restore the encoded logical state.

**Rotated surface code and memory experiment**. The rotated surface code (Fowler et al., 2012) is a specific stabilizer code that encodes a single logical qubit in a two-dimensional lattice of physical qubits. It is currently the only QEC architecture integrated into full-stack quantum control systems, where real-time decoding poses an immediate engineering constraint rather than a purely theoretical concern. As shown in Fig. 1, a distance-$d$ rotated surface code is a $[\![d^2, 1, d]\!]$ stabilizer code, comprising $d^2$ data qubits located at the vertices, interleaved with $d^2 - 1$ ancilla qubits used to measure the $X$ and $Z$ stabilizer operators and and extract the error syndrome $\boldsymbol{s} \in \{0,1\}^{d^2-1}$.

As a concrete instantiation of the rotated surface code, the memory experiment represents the most basic form of QEC, while already capturing the core challenges of fault-tolerant computation (AI, 2023). Following conventions (AI, 2023; 2025), we focus on the $Z$-memory experiment. In this context, we initialize a known single logical qubit state $|0\rangle_L$ encoded into $d^2$ data qubits, perform $r$ rounds of syndrome measurements (typically $r = d$), and finally measure the resulting state by the logical operator $\bar{Z}_L$, as a Pauli string $Z^{\otimes d}$ supported along a minimal path that spans the lattice. Given a spatiotemporal syndrome volume $\boldsymbol{s} \in \{0,1\}^{r \times (d^2-1)}$, the decoder $f$ must utilize this dense 3D history to infer the cumulative logical update, strictly approximating the maximum likelihood inference:

$$f : \boldsymbol{s} \in \{0,1\}^{r \times (d^2-1)} \to \hat{L} \in \{I, X_L, Y_L, Z_L\}. \quad (1)$$

The inferred logical update $\hat{L}$ specifies how the final logical state differs from the initial encoding. In this context, the memory experiment is deemed successful if $\hat{L}$ matches the initially encoded logical operator (i.e., $\hat{L} \in \{I, \bar{Z}_L\}$).

**QEC decoding as binary classification**.

Based on the formulation of the $Z$-memory experiment in Eq. (1), the QEC decoding task can be naturally cast as a binary classification problem, where the label $y \in \{0,1\}$ indicates whether the inferred logical update $\hat{L}$ corresponds to a successful or failed decoding outcome. Mathematically, the neural decoder (or binary classifier) aims to estimate the conditional probability of $\hat{L}$ given the observed syndrome $\boldsymbol{s}$, i.e.,

$$P(\hat{L}|\boldsymbol{s}) = f(\boldsymbol{s}; \boldsymbol{\theta}), \quad (2)$$

where the neural network is parameterized by $\boldsymbol{\theta}$. For example, by minimizing the binary cross-entropy loss against the ground truth $y$, the neural decoder learns to identify complex, correlated error patterns that signal a logical failure.

**Related works**. Prior neural decoders primarily emphasize architectural innovations to improve decoding accuracy. In contrast, our work focuses on a systematic analysis under accuracy–latency constraints, resulting in limited overlap in scope. Refer to Appendix B for more details.

## 3. Neural Decoders and Compression Pipeline

This section presents our framework for analyzing neural decoders under the accuracy–efficiency tension. Specifically, to address Q1, Sec. 3.1 introduces a taxonomy of neural architectures based on inductive biases toward surface code topology. To answer Q2, Sec. 3.2 presents an end-to-end compression pipeline that produces lightweight neural decoders under real-time constraints, and Sec. 3.3 evaluates the resulting models through hardware resource estimation to assess their deployability on FPGA platforms.

### 3.1. Implementation of Neural Decoder Architectures

To enable a systematic evaluation of neural decoder designs, i.e., $f(\boldsymbol{s}; \boldsymbol{\theta})$ in Eq. (2), we curate a portfolio of five representative architectures that span the dominant inductive biases explored in modern supervised decoding. Rather than adopting prior implementations verbatim, we systematically redesign these paradigms to explicitly expose and satisfy the accuracy–efficiency requirements imposed by fault-tolerant quantum operation and real-time FPGA deployment. Below, we describe the implementation details of each neural decoder, with additional details deferred to Appendix C.

**Multilayer perceptron (MLP).** We implement the MLP as a structure-agnostic baseline with minimal inductive bias. Treating the syndrome volume $\boldsymbol{s}$ in Eq. (1) as a flattened vector, MLP consists of dense affine transformations:

$$f_{\text{MLP}}(\boldsymbol{s}) = \sigma(\mathbf{W}_L \cdots \sigma(\mathbf{W}_1 \text{vec}(\boldsymbol{s}) + \mathbf{b}_1) \cdots + \mathbf{b}_L), \quad (3)$$

where $\boldsymbol{\theta} = (\mathbf{W}_l, \mathbf{b}_l)_l$ and $\sigma(\cdot)$ is the activation function. While universally expressive, MLP discards all spatiotemporal locality, serving primarily to benchmark the value of the geometric priors introduced in subsequent models.

**Dilated 3D-Convolutional neural network (CNN).** This widely adopted architecture employs translation invariance to naturally align with the surface code lattice (Varsamopoulos et al., 2017; Torlai & Melko, 2017; Chamberland & Ronagh, 2018). A critical deviation in our design is the strict exclusion of pooling layers, which are ubiquitous in standard computer vision but detrimental in QEC as they degrade the spatial resolution required to pinpoint error locations. Instead, to expand the receptive field effectively, we employ dilated convolutions (Yu & Koltun, 2015), i.e.,

$$f_{\text{CNN}}(\boldsymbol{s}) = \text{Head}(\mathbf{K} *_{\delta_L} \cdots (\mathbf{K} *_{\delta_2} (\mathbf{K} *_{\delta_1} \boldsymbol{s}))), \quad (4)$$

where $(\mathbf{K} *_{\delta_l} \boldsymbol{s})[x,y,t] = \sum_{i,j,k} \mathbf{K}[i,j,k] \cdot \boldsymbol{s}[x + i\delta_l, y + j\delta_l, t + k\delta_l]$, $\mathbf{K}[i,j,k]$ denotes the $3 \times 3 \times 3$ kernel with $i,j,k \in \{-1,0,1\}$, $*_{\delta_l}$ refers to the dilated convolution with dilation rate $\delta_l \in \mathbb{Z}^+$, and $\text{Head}(\cdot)$ is a final fully connected classifier. Remarkably, this strategy preserves the full spatiotemporal dimensionality of the syndrome volume $\boldsymbol{s}$, a design choice motivated by the locality of stabilizer codes and supported by AlphaQubit (Bausch et al., 2024).

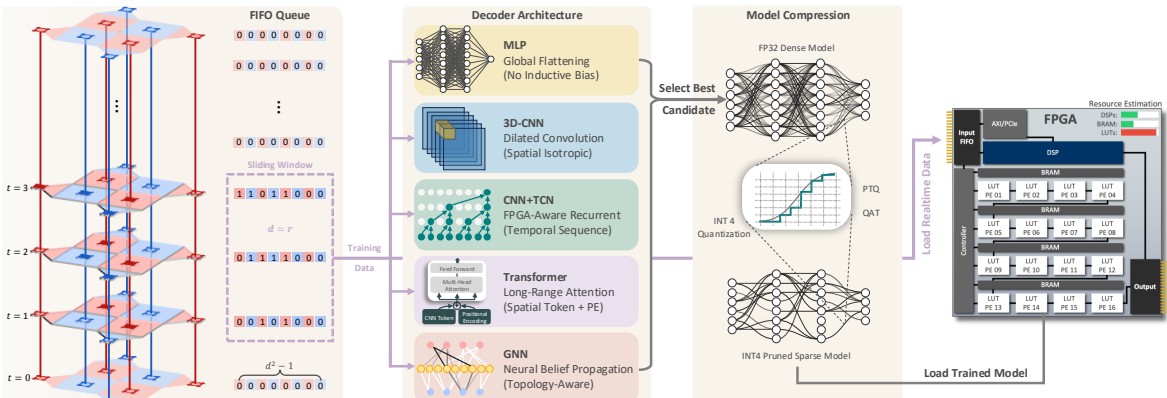

*Figure 2.* **Schematic illustration of the entire pipeline. Left:** Syndrome data is processed via a causal sliding window ($r = d$) to maintain fault-tolerant boundaries. **Middle:** A diverse spectrum of neural decoders is evaluated, ranging from MLPs and 3D-CNNs to temporal models, Transformers, and topology-aware GNNs. **Right:** To resolve the latency bottleneck, we employ an aggressive INT4 compression pipeline (combining Pruning and QAT) to map these models onto a resource-constrained FPGA accelerator. As illustrated in the hardware micro-architecture, the Processing Element array is constructed primarily using LUTs to offload computation from scarce DSP resources, ensuring physical feasibility on standard FPGA fabrics.

**Temporal convolutional networks (TCN).** To process the time-domain sequence while mitigating the cubic complexity of 3D-CNNs in Eq. (4), we adopt a spatially decoupled architecture ($f_{\text{time}} \circ f_{\text{space}}$). While prior works predominantly utilize LSTMs or GRUs for the temporal component (Baireuther et al., 2018; Chamberland & Ronagh, 2018; Varbanov et al., 2025), we explicitly opt for a TCN, considering the critical hardware constraint. That is, standard RNNs rely on recursive state updates and sensitive gating functions (sigmoid/tanh), which suffer from saturation and degradation under low-bit quantization. In contrast, our TCN implementation consists solely of 1D and 2D convolutions with ReLUs, i.e.,

$$f_{\text{TCN}}(\boldsymbol{s}) = \text{Head}(\text{Conv1D}([g(\boldsymbol{s}_1), \cdots, g(\boldsymbol{s}_r)])), \quad (5)$$

where the function $g(\boldsymbol{s}_i) = \prod_{l=1}^{L} \text{Conv2D}_l(\boldsymbol{s}_i)$ with $i \in [r]$ and $\boldsymbol{s}_i \in \{0,1\}^{(d^2-1)}$ refers that the syndrome data $\boldsymbol{s}_i$ at the $i$-th round is mapped onto a 2D spatial grid via a constant embedding. The operation $\text{Conv2D}_l(\cdot)$ denotes the $l$-th 2D convolutional layer that processes each round of syndrome data independently. As will be shown in Sec. 4, this structural homogeneity ensures robustness and enables fully parallel, non-autoregressive processing across the temporal dimension of $\boldsymbol{s}$.

**Transformer.** To capture long-range correlations of $\boldsymbol{s}$, a crucial research line is developing Transformer-based neural decoders, such as AlphaQubit series (Bausch et al., 2024; Senior et al., 2025). However, we introduce a substantial modification to the input stage to bridge the gap between experimental and simulation environments. Unlike hardware experiments in Ref. (Bausch et al., 2024) where input vectors contain rich analog information (e.g., soft readout probabilities), standard simulations provide only sparse binary syndromes. A direct linear projection of these sparse bits yields degenerate embeddings. To resolve this, we engi-

neer a convolutional tokenizer ($\mathcal{T}$) combined with explicit positional encoding ($\mathbf{P}$), i.e.,

$$f_{\text{Trans}}(\boldsymbol{s}) = \text{Head}(\text{Attention}(\mathcal{T}(\boldsymbol{s}) + \mathbf{P})). \quad (6)$$

Here, the tokenizer $\mathcal{T}(\cdot)$ aggregates local binary patches into dense feature vectors before they enter the attention mechanism $\text{Attention}(\cdot)$. It is worth noting that while this implementation tailors for a fixed spatiotemporal window ($r = d$), it conceptually functions as the transformer block of the AlphaQubit (Bausch et al., 2024). To scale to continuous decoding with indefinite rounds ($r \gg d$), this module can be explicitly wrapped in a recurrent loop: by feeding the latent state and decoding output of the current window into the input of the next iteration, one can fully reconstruct the recurrent architecture required for streaming decoding.

**Graph neural network (GNN).** GNNs provide a versatile framework for message passing and feature aggregation on high-dimensional, non-planar structures. Consequently, they have been employed for QEC decoding on both detector graphs (Lange et al., 2025) and Tanner graphs (Liu & Poulin, 2019). In this work, however, we prioritize GNN formulations on the Tanner graph (detailed in Appendix B). In contrast to prior neural decoders that perform label-level prediction, GNN-based decoders make predictions at the detector level, i.e., $f_{\text{GNN}} : \boldsymbol{s} \rightarrow \hat{\mathbf{p}} \in [0,1]^{|\mathcal{V}_d|}$. To apply GNNs, the surface code is first expressed as a Tanner graph $\mathcal{G} = (\mathcal{V}_c \cup \mathcal{V}_d, \mathcal{E})$, where $\mathcal{V}_c$ and $\mathcal{V}_d$ separately denote detector nodes and data-qubit nodes, and edges $\mathcal{E}$ encode local stabilizer constraints.

Operating on this representation, the GNN approximates maximum-likelihood decoding via neural belief propagation (Liu & Poulin, 2019). Unlike standard belief propagation, whose fixed update rules suffer from message oscillations on surface codes due to short cycles, the GNN

employs learnable message-passing functions to mitigate these instabilities.

Formally, the iterative update of the node $v$ at the $k$-th step is $\mathbf{h}_v^{(k)} = \text{GRU}\big(\mathbf{h}_v^{(k-1)}, \bigoplus_{u \in \mathcal{N}(v)} \phi(\mathbf{h}_u^{(k-1)}, \mathbf{e}_{uv})\big)$, where $\mathcal{N}(v)$ is the set of its neighbors in the Tanner graph $\mathcal{G}$, $\bigoplus$ represents a permutation-invariant aggregation, the learnable function $\phi(\cdot, \cdot)$ computes messages along edges $\mathbf{e}_{uv} \in \mathcal{E}$, and GRU integrates the aggregated messages. Consequently, the GNN-based neural decoder is expressed as the readout of the data node embeddings after $K$ iterations:

$$f_{\text{GNN}}(\boldsymbol{s}) = \sigma\left(\text{Readout}(\mathbf{h}_v^{(K)})\right), \qquad (7)$$

where $\sigma(\cdot)$ maps the latent states to the prediction $\hat{\mathbf{p}}$.

### 3.2. Model Compression Pipeline

To make large and computationally intensive neural decoders deployable on resource-constrained FPGAs, we adopt an aggressive compression pipeline that sequentially applies weight quantization and pruning (Bausch et al., 2024), as shown in Fig. 2. By targeting extreme sparsity and low-bit precision (down to INT4), our approach achieves substantial reductions in memory footprint and logic utilization without sacrificing decoding fidelity.

**Quantization formalism.** We adopt a uniform symmetric quantization scheme (Jacob et al., 2018). For a given tensor $\boldsymbol{x}$ (weights or activations in neural networks), the quantized integer representation $\boldsymbol{x}_{\text{int}}$ is given by

$$\boldsymbol{x}_{\text{int}} = \text{clamp}\left(\left\lfloor\frac{\boldsymbol{x}}{\eta}\right\rceil, -2^{b-1}, 2^{b-1}-1\right) \cdot \eta \qquad (8)$$

where $b$ is the bit-width (e.g., $b = 4$ for INT4), $\eta$ is a learnable or calibrated scale factor, and $\lfloor \cdot \rceil$ denotes the rounding-to-nearest operation. To maximize representation fidelity, we employ per-channel scaling for weights to accommodate varying dynamic ranges across output filters, and per-tensor scaling for activations to minimize hardware overhead.

The proposed pipeline in Fig. 2 begins with post-training quantization (PTQ) (Nagel et al., 2020) as a feasibility probe. PTQ alone typically faces a "cliff effect" when tackling low-bit quantization with non-negligible accuracy degradation. To recover this loss, the proposed pipeline further employs quantization aware training (QAT) (Jacob et al., 2018) to insert differentiable fake-quantization nodes in Eq. (8) into the forward pass of the FP32 seed model. Since the rounding operation has zero gradient almost everywhere, we utilize the straight-through estimator (Jacob et al., 2018) during back-propagation, i.e., $\frac{\partial \mathcal{L}}{\partial \boldsymbol{x}} \approx \frac{\partial \mathcal{L}}{\partial \boldsymbol{x}_q}$. This approximation allows the optimizer to update the latent floating-point weights to robust minima that are resilient to quantization noise. We implement this pipeline using Brevitas (Pappalardo et al., 2025), initializing QAT with FP32 model parameters to accelerate convergence.

**Sparse topology exploration via pruning.** The second compression stage is the unstructured magnitude pruning, applied after the quantization constraints are established. This sequence is necessary, ensuring that the pruning criterion operates on the effective deployable values rather than latent floating-point weights. The algorithm implementation is as follows. Given the quantized weight tensor $\mathbf{W}_q$, a binary mask $\mathbf{M} \in \{0, 1\}^{|\mathbf{W}_q|}$ is generated based on the magnitude of the integer representations, i.e.,

$$\mathbf{M}_i = \mathbb{I}(|\mathbf{W}_q[l, i]| > \tau_k), \qquad (9)$$

where $\mathbf{W}_q[l, i]$ denotes the $i$-th weight on the $l$-th layer, $\mathbb{I}(\cdot)$ is the indicator function and $\tau_k$ is the threshold determined by the target sparsity level $k$. To mitigate the accuracy loss introduced by aggressive connectivity reduction induced by high-sparsity pruning, we conduct a brief phase of sparsity-aware fine-tuning, enabling the remaining active weights to adapt to the reduced connectivity.

**Remark**. While standard dense accelerators cannot easily exploit unstructured sparsity (Han et al., 2016), FPGA synthesis flows offer a unique advantage: static zero-valued weights allow the synthesis tool to optimize away the corresponding logic gates (logic trimming), reducing the effective LUT utilization (see Sec. 3.3). This post-quantization pruning serves as a fine-grained resource scaling knob, probing the minimal parameter count required to sustain logical error suppression under strict bit constraints.

### 3.3. FPGA Resource Estimation

To rigorously assess the feasibility of deploying neural decoders on FPGAs under latency constraints, we develop a resource estimation model. For completeness, we begin by briefly reviewing FPGA fundamentals and deployment considerations, followed by the estimation framework.

**Foundations of FPGAs**. Modern FPGA architectures consist of three primary resource types. As shown in Fig. 2, these include Block RAM (BRAM) for on-chip storage of weights and activations, DSP slices for high-precision arithmetic operations such as FP32/INT8, and Look-Up Tables (LUTs) for general-purpose logic and low-bit computation.

Since LUTs are orders of magnitude more abundant than DSP slices, it is preferable to map neural decoders to LUT-based implementations whenever possible. This motivates the use of INT4 arithmetic, which allows multiplication to be synthesized using LUTs rather than DSPs. As a result, the quantized neural decoders in Sec. 3.2 are a prerequisite for transitioning FPGA deployment from a *DSP-bound* to a *logic-bound* regime and achieving maximal efficiency.

**FPGA resource mapping for neural decoders.** Under the LUT-oriented deployment strategy established above, FPGA feasibility is evaluated by expressing the neural decoder in terms of its underlying arithmetic operations. To this end,

we decompose the optimized neural decoder into elementary INT4 multiply–accumulate (MAC) operations, noting that each linear operation in the network corresponds to a fixed number of MACs. Summing over all layers yields the total number of INT4 MACs required for one inference pass, which quantifies the computational workload of the decoder.

To map this workload onto FPGA hardware, we adopt the standard notion of a processing element (PE) as a minimal compute unit that executes one INT4 MAC per clock cycle. The number of available PEs thus determines the maximum degree of parallel MAC execution on the FPGA. Based on synthesis results for Xilinx UltraScale+ devices, implementing one such PE consumes approximately $\beta \approx 20$ LUTs. According to $\beta$ and the total number of available LUTs, we can directly determine both the achievable degree of parallel MAC execution and the resulting inference latency for a given neural decoder on an FPGA.

**Latency estimation protocol.** We now present the proposed resource estimation model, consisting of three key steps. First, we compute the maximum number of PEs supported by a given FPGA, i.e., $P_{\max} = N_{\mathrm{LUT}}/\beta$ with $N_{\mathrm{LUT}}$ being the total available LUTs, which sets the degree of parallel MAC execution per cycle. Second, since neural network layers are executed sequentially, we calculate the latency of each layer. In particular, for the $l$-th layer with a workload of $N_l$ MAC operations, the required execution cycles are given by $\lceil N_l/P_{\max} \rceil$. Last, the total cycles is obtained by taking the summation over all $L$ layers $T_{\mathrm{cc}} \approx (1+\gamma) \cdot \sum_{l=1}^{L} \left\lceil \frac{N_l}{P_{\max}} \right\rceil$, where $\gamma$ is a safety overhead factor (set to 10%) to account for control logic bubbles and inter-layer synchronization. The wall-clock latency yields $t = T_{\mathrm{cc}}/f_{\mathrm{clk}}$ with $f_{\mathrm{clk}}$ being the FPGA frequency.

**Remark.** The above analytical model adopts a conservative 50% derating factor (without full Vivado place-and-route flow), and therefore does not model routing congestion or timing closure. Appendix D.4 complements this with Vitis HLS synthesis on VP1902: in the case of $d = 9$, TCN meets the microsecond deadline and is 14% faster than our analytical estimate, indicating the model is conservative rather than optimistic.

## 4. Experiment Results

In this section, we address Q1 and Q2 by systematically evaluating QEC decoders' performance on the rotated surface code with distances $d \in \{3, 5, 7, 9\}$. Following conventions, we fix the syndrome measurement rounds as $r = d$ and focus on the $Z$-memory experiment introduced in Sec. 2. The datasets are collected using Stim (Gidney, 2021) under a standard circuit-level depolarizing noise model with an error rate $p = 0.005$ (near the pseudo-threshold). We use the logical error rate (LER) as the primary metric and benchmark five neural decoders in Sec. 3.1 against

*Table 1.* **Impact of noise priors on Google Sycamore hardware data.** Performance comparison of TCN-small against standard and correlated MWPM. The neural decoder is evaluated in two regimes: *Zero-shot* (directly applied after pretraining) and *Finetuned* (FT). "Uniform Prior" denotes pretraining on synthetic depolarizing noise ($p = 0.005$), while "Calibrated Prior" utilizes Google's device-specific error model. Values denote logical error rate (LER, %).

| $d$ | Noise Prior / Model | MWPM Baselines | | Neural Decoder (TCN) | |
|---|---|---|---|---|---|
| | | Standard | Correlated | Zero-shot | Finetuned |
| 3 | Uniform ($p$=0.005) | 8.01 | 7.38 | 34.42 | 9.27 |
| | Calibrated Data | | | **6.81** | **6.70** |
| 5 | Uniform ($p$=0.005) | 14.38 | 12.52 | 47.89 | 20.06 |
| | Calibrated Data | | | **11.59** | **11.47** |

the MWPM decoder, i.e., PyMatching v2 (Higgott & Gidney, 2025). Detailed hyperparameters and platform specifications are explained in Appendix D.1. Source code is available at https://github.com/GrahamYan/QEC-neural-decoder-benchmark.

### 4.1. Neural Decoder Benchmarking

Our first task is to *determine the optimal operating regime for neural decoding*. To this end, we conduct a scaling analysis across five representative architectures. We systematically vary the code distance with $d \in \{3, 5, 7, 9\}$ and training dataset size from $10^5$ to $2.5 \times 10^7$ samples. For each decoder, we consider two model scales ("Small" and "Large") to examine the effect of model capacity.

**Decoding performance is primarily driven by data scale with appropriate inductive bias**. The achieved results are visualized in Fig. 3(a). That is, among architectures whose inductive biases are aligned with the spatiotemporal structure of the surface code (e.g., CNNs, Transformers, and TCNs), decoding accuracy converges once the training data exceeds $10^7$ samples. This suggests that QEC decoding operates in a **data-first regime**: given an architecture with appropriate inductive bias aligned to the problem geometry, data scale becomes the primary driver of performance.

Crucially, our analysis of parameter scaling highlights the necessity of efficient capacity expansion. As shown in Table 9, the standard 3D-CNN baseline exhibits varying parameter growth, which balloons to 32.5 million at $d = 9$ due to its dense classifier scaling with cubic volume. In stark contrast, other temporal architectures (TCN, Transformer) maintain a highly compact footprint. Even in their "Large" configurations, they contain fewer than 1 million parameters (see Table 2). This demonstrates that massive parameterization is not a prerequisite for high accuracy; rather, compact models with efficient inductive biases are sufficient to solve the decoding problem, provided they are supported by adequate data scale.

**Real-world data validation.** We validate the above findings on real quantum hardware using experimental data

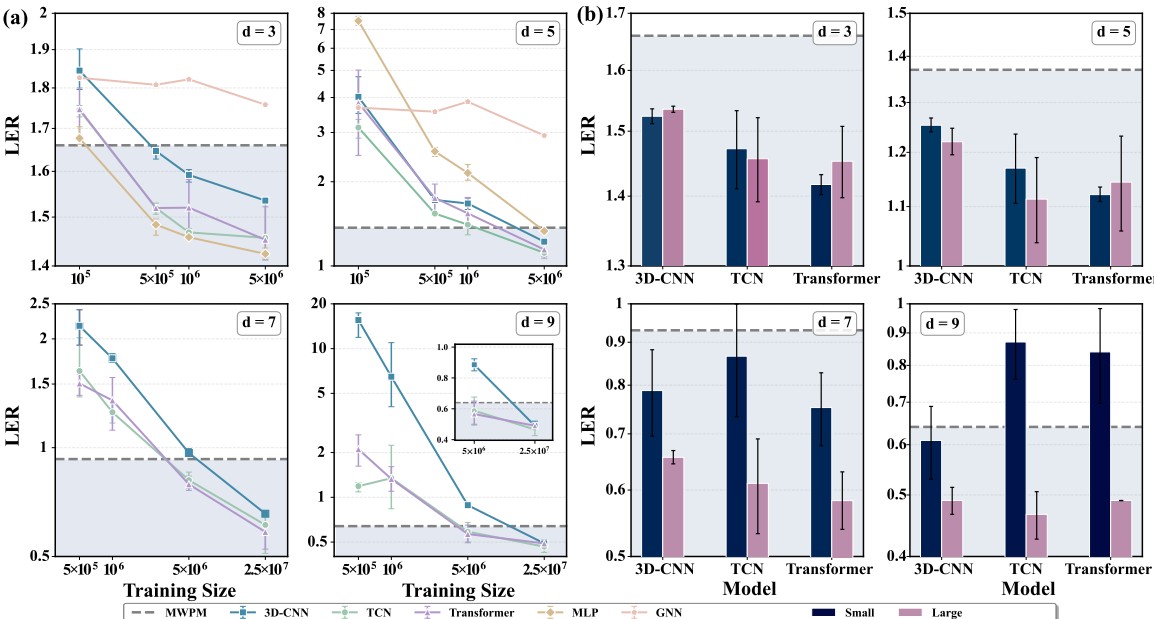

*Figure 3.* **Neural decoder benchmark on the rotated surface code.** **(a)** LER vs. training set size with model size "large". The dashed gray line marks the MWPM baseline; the shaded region indicates LER below MWPM. Error bars show the min–max range over three independent runs. Inset ($d$=9): 3D-CNN outliers at $5 \times 10^5$ and $10^6$ samples (15.6% and 6.5%) are shown separately to preserve the main axis scale. **(b)** Model comparison across width ratios at the largest available training size ($d$=3, 5: $5 \times 10^6$; $d$=7, 9: $2.5 \times 10^7$). Bar heights indicate mean LER; error bars show the min-max range.

from the Google Sycamore processor ($d = 3, 5$). We employ a compact and small TCN decoder in Eq. (5) trained on $5 \times 10^6$ samples and investigate the impact of noise model alignment. For reference models, standard MWPM (assuming independent noise) and correlated MWPM (aware of experimental bias correlations) are adopted. Besides, two pretraining strategies are employed for the TCN decoder, i.e., (1) Uniform Prior, where the model is trained on standard depolarizing noise ($p = 0.005$), and (2) Calibrated Prior, where the training data is generated using a noise model derived from device calibration.

The results in Table 1 demonstrate that accurate error characterization is the decisive factor for decoding performance. With a calibrated prior, the neural decoder (zero-shot) significantly outperforms standard MWPM and rivals or exceeds correlated MWPM even without fine-tuning. While fine-tuning further suppresses LER, the substantial gain from switching priors underscores that the primary advantage of neural decoders lies in their capacity to internalize complex, non-Pauli, and correlated error mechanisms that are intractable for rigid graph-based heuristics.

Crucially, these experimental results bridge the gap between our simulation benchmarks and practical deployment. They validate that the lightweight architectures proposed in Section 3 are not confined to synthetic environments but are fully capable of handling real-world noise when informed by device calibration data. Moreover, we observe that the performance advantage of the neural decoder over MWPM

is even more pronounced on physical hardware than in idealized simulations. This suggests that while classical decoders struggle with the complex, non-Pauli correlations (e.g., crosstalk, leakage) inherent in superconducting processors, neural decoders effectively capture these latent error patterns. Consequently, deploying such neural decoders offers a distinct strategic advantage for maximizing the error suppression capabilities of current devices.

### 4.2. Model Compression and Optimization

Our second task is to explore the effectiveness of the proposed compression pipeline. Based on the scaling laws established in Sec. 4.1, we select the most hardware-efficient checkpoints of the neural decoders as our baselines. These decoders rely on FP32 accuracy headroom to absorb compression-induced fidelity loss and preserve an LER below the MWPM baseline.

**QAT is necessary in neural decoding**. Fig. 4(a) reveals a striking disparity in quantization resilience across architectures. While the spatial 3D-CNN remains robust under standard PTQ at W8A8 (where W and A stand for weights and activations, respectively), the temporal architectures (TCN, Transformer) suffer catastrophic degradation even at 8-bit precision, yielding LERs orders of magnitude worse than MWPM. This fragility stems from the cumulative precision errors in sequential processing stages, rendering standard PTQ insufficient for advanced sequence models.

Furthermore, at the target W4A4 precision, all architectures

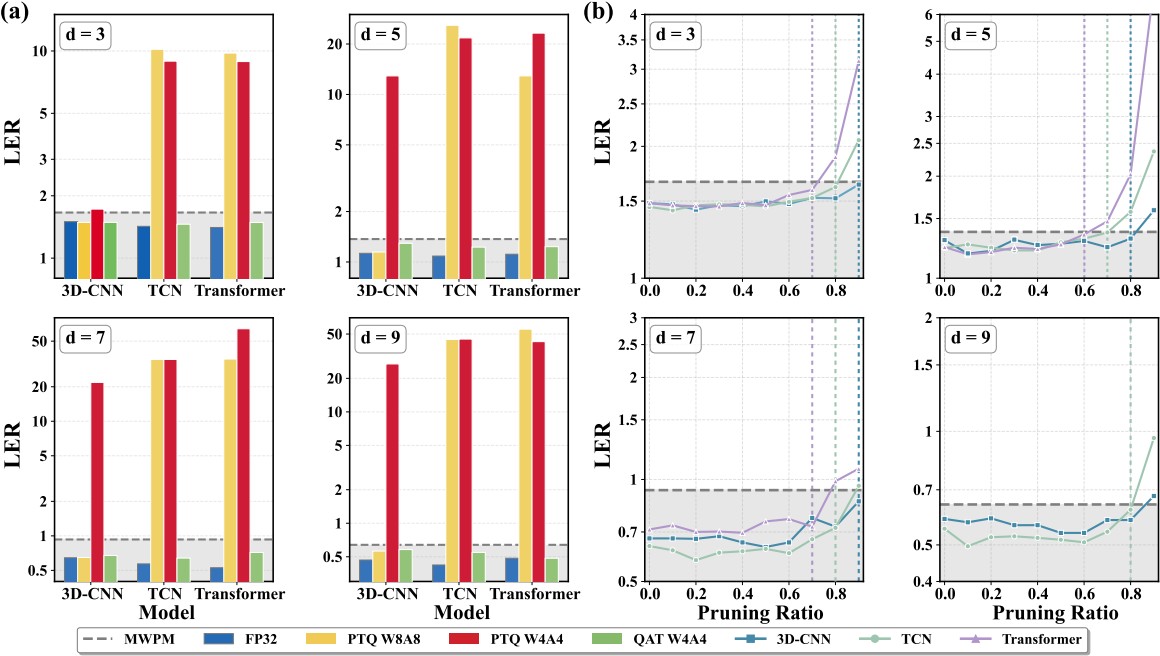

*Figure 4.* **Impact of quantization and pruning on decoder fidelity. (a)** Quantization sensitivity under PTQ and QAT. While 3D-CNN retains robustness at W8A8, TCN and Transformer exhibit severe degradation even at 8-bit precision, and all models fail at W4A4. However, QAT is able to recover the accuracy loss at INT4. **(b)** Pruning robustness based on the recovered QAT-W4A4 models. The curves track LER as unstructured sparsity increases. Vertical dotted lines mark the "breakdown point" where performance deteriorates to the MWPM level.

collapse under PTQ. Consequently, we transition to QAT, and QAT successfully recovers the accuracy for all architectures at W4A4, consistently suppressing LER below the MWPM threshold. This result validates that QAT is not merely an optimization for low-bit regimes, but a fundamental prerequisite for deploying high-performance temporal decoders.

**Sparsity-accuracy trade-off via pruning.** Building upon the robust QAT-W4A4 checkpoints, we further reduce logic utilization via unstructured pruning. Fig. 4(b) illustrates the trajectory of LER as sparsity increases. We identify a critical "breakdown point" for each model-the maximum sparsity ratio before the decoder's error rate exceeds the MWPM baseline (dashed line). For instance, the TCN and 3D-CNN demonstrate remarkable resilience, sustaining fault-tolerant performance up to $60\% \sim 70\%$ sparsity across code distances. These empirically determined limits define the final sparse INT4 models used for the hardware resource projection in the following section.

### 4.3. Resource Estimation

Building upon the compressed INT4 neural decodes in Sec. 4.2, our third task is to evaluate their deployability on two representative FPGA platforms: the VP1802 with 1.68M available LUTs and $P_{max} = 84,022$; and VP1902 with 4.23M available LUTs and $P_{max} = 211,507$. A comprehensive justification for targeting these specific architectures, along with extended hardware specifications, is

*Table 2.* **Clock cycle (CC) estimation for neural decoders.** Pruning ratios are selected as the maximum sparsity preserving sub-MWPM performance (Fig. 4b).

| Model | $d$ | Scale | Prune | MACs | Size | Estimated CC | |
|---|---|---|---|---|---|---|---|
| | | | | | | VP1802 | VP1902 |
| 3D-CNN | 3 | Small | 90% | 721K | 7 KB | 20 | 13 |
| | 5 | Small | 80% | 4.9M | 47 KB | 73 | 33 |
| | 7 | Large | 90% | 23.7M | 432 KB | 317 | 129 |
| | 9 | Large | 80% | 95.7M | 3.2 MB | 1,261 | 505 |
| TCN | 3 | Small | 80% | 983K | 10 KB | 20 | 10 |
| | 5 | Small | 70% | 5.5M | 17 KB | 75 | 35 |
| | 7 | Large | 80% | 36.7M | 51 KB | 486 | 195 |
| | 9 | Large | 80% | 59.0M | 63 KB | 779 | 314 |
| Transformer | 3 | Small | 70% | 2.3M | 32 KB | 42 | 25 |
| | 5 | Small | 60% | 11.6M | 43 KB | 164 | 73 |
| | 7 | Large | 70% | 84.8M | 127 KB | 1,119 | 454 |

provided in Appendix D.3. Using the resource estimation methodology established in Section 3.3, we assess the required $T_{cc}$ and latency for each configuration.

We assume a nominal operating frequency of $f_{clk} = 300$MHz, where a budget of 300 clock cycles corresponds to 1 $\mu$s. The analyzed results in Table 2 reveal **three distinct feasibility regimes** for the compressed neural decoders.

**Effortless regime for** $d = 3, 5$. For near-term code distances, hardware latency is negligible. All architectures complete decoding within tens of cycles (e.g., $T_{cc} < 80$ CCs), leaving ample headroom for control logic overhead.

**Transition regime** $d = 7$. As complexity scales, the Transformer architecture begins to exceed the microsecond budget on the smaller VP1802 ($T_{cc} = 1,119$ CCs). The 3D-CNN remains within the budget on VP1902 ($T_{cc} = 129$ CCs), while the TCN also stays feasible on VP1902 ($T_{cc} = 195$ CCs). With standard compilation optimizations (e.g., pipelining), these models can comfortably meet real-time constraints on commercial-grade FPGAs.

**Critical regime** ($d = 9$). A sharp divergence occurs at this scale. Most architectures breach the latency barrier, rendering them impractical for real-time feedback. Notably, only the TCN demonstrates continued viability on the high-end VP1902, with an analytical estimate of $T_{cc} = 314$ CCs. HLS synthesis via Vitis HLS further confirms this feasibility, reporting a latency of 271 CCs ($0.77\,\mu s$ at 350 MHz) for the same configuration (see Appendix D.4). This demonstrates that it is feasible to deploy a neural decoder that surpasses MWPM accuracy at $d = 9$ well within the strict $1\,\mu s$ timing window on a cutting-edge FPGA.

## 5. Conclusion

In this study, we conducted a systematic study of neural decoding for surface codes under explicit accuracy–latency constraints. By unifying representative architectures and enforcing hardware-aware compression, we showed that neural decoding performance is driven more by data scale and inductive bias than architectural complexity. Moreover, we uncover that quantization-aware training is necessary to meet microsecond-scale latency on FPGAs. All of these findings provide concrete guidance for scalable, real-time, hardware–algorithm co-designed neural QEC decoding.

## Impact Statement

This paper presents work whose goal is to advance the field of Machine Learning for quantum computing, especially quantum error correction. There are many potential societal consequences of our work, none which we feel must be specifically highlighted here.

## Acknowledgements

YXD acknowledges the funding from A*STAR (R25J4IR109) and NTU SUG (025257-00001).

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

# A. Extended Preliminaries

## A.1. Detailed Definition of QEC, Rotated Surface Code, and Memory Experiments

Consider a system of $n$ physical qubits with Hilbert space $\mathcal{H} = (\mathbb{C}^2)^{\otimes n}$. Let $\mathcal{P}_n = \{\pm 1, \pm i\} \times \{I, X, Y, Z\}^{\otimes n}$ denote the $n$-qubit Pauli group. A stabilizer code $[\![n, k, d]\!]$ is defined by an abelian subgroup $\mathcal{G} \subset \mathcal{P}_n$ such that $-I \notin \mathcal{G}$. The code space $\mathcal{C}$ is the $+1$ eigenspace of $\mathcal{G}$:

$$\mathcal{C} = \{|\psi\rangle \in \mathcal{H} : G|\psi\rangle = |\psi\rangle, \forall G \in \mathcal{G}\}.$$

The group $\mathcal{G}$ is typically specified by $m = n - k$ independent generators, $\mathcal{G} = \langle g_1, \ldots, g_m \rangle$. For stabilizer codes, these generators can be partitioned into $X$-type and $Z$-type operators, $\mathcal{G} = \langle \mathcal{G}_X, \mathcal{G}_Z \rangle$. The code distance $d$ is defined as the minimum weight of an operator in $N(\mathcal{G}) \setminus \mathcal{G}$ (i.e., the minimum weight of a non-trivial logical operator). Such a code can correct any arbitrary error with weight up to $\lfloor (d-1)/2 \rfloor$.

An error $E \in \mathcal{P}_n$ moves the state out of $\mathcal{C}$ if it anticommutes with any generator. We define the syndrome extraction map $\sigma : \mathcal{P}_n \to \mathbb{F}_2^m$, mapping a Pauli error to a binary syndrome vector. The $i$-th component of the syndrome $s = \sigma(E)$ is determined by the commutation relations:

$$g_i E = (-1)^{s_i} E g_i, \quad s_i \in \{0, 1\}.$$

Two errors $E$ and $E'$ share the same syndrome if and only if $E^\dagger E' \in \mathcal{G}$. Such errors are termed degenerate and act identically on the logical subspace. The set of logical operators coincides with the normalizer of $\mathcal{G}$ in $\mathcal{P}_n$, denoted by $N(\mathcal{G}) = \{P \in \mathcal{P}_n : PGP^\dagger = G, \forall G \in \mathcal{G}\}$. The effective logical Pauli group is given by the quotient group $\mathcal{L} = N(\mathcal{G})/\mathcal{G}$. The decoding objective is to infer the most probable logical error class given the observed syndrome $s$. Let $P(E)$ be the physical error probability distribution (e.g., i.i.d. depolarizing noise). The decoder seeks to maximize the logical frame probability:

$$\hat{L} = \underset{L \in \mathcal{L}}{\operatorname{argmax}} \sum_{G \in \mathcal{G}} P(E_{rec} \cdot L \cdot G \mid s),$$

where $E_{rec}$ is a fixed canonical recovery operator satisfying $\sigma(E_{rec}) = s$. This formulation casts decoding as a coset classification problem.

**Memory experiments**. Exploiting the Calderbank-Shor-Steane (CSS) structure of surface codes, $X$-type and $Z$-type errors can be detected and corrected independently. Consequently, without loss of generality, this study focuses on correcting $X$ errors (which cause logical bit-flips) within a Z-memory experiment as in AI (2023).

## A.2. Superconducting Clock Cycle, the SI1000 Model, and Microsecond-scale Decoding Latency

The imposition of a strict *microsecond-scale decoding latency* is derived from the operational characteristics of transmon-based superconducting processors. In this subsection, we provide a brief explanation for completeness.

Following the definitions in recent experimental benchmarks (Gidney et al., 2021) (e.g., the Google Quantum AI "Honeycomb" memory), a standard QEC cycle is composed of reset, gate operations, and measurement. This regime is formally encapsulated by the SI1000 (Superconducting-Inspired, 1000 ns) noise model. In a typical SI1000 cycle, the timeline is dominated not by unitary gates (which are fast, $\approx$ 20-40 ns), but by the readout and reset operations, which typically span 500-800 ns to ensure high fidelity and signal depletion. As detailed in Table 3, the summation of these operations necessitates a cycle time $t_{\text{cycle}} \approx 1\,\mu s$. Since the decoder must infer corrections for round $t$ before the measurement outcomes of round $t+1$ overwrite the buffer (or to perform active feedback), the decoding latency is strictly upper-bounded by this duration.

*Table 3.* **Breakdown of a single QEC cycle** ($1\,\mu s$). Timing estimates are based on the SI1000 model for transmon qubits (Gidney et al., 2021). The cycle consists of 8 parallel layers. Note that measurement and reset dominate the timeline, providing the hard real-time constraint for the decoder.

| Layer | Operation | Duration (ns) | Cumulative (ns) |
|---|---|---|---|
| 1 | Qubit Reset / Initialization | 200 | 200 |
| 2 | Single-qubit Gates ($H$, etc.) | 25 | 225 |
| 3–6 | Two-qubit Gates (CNOT/CZ) | $4 \times 30$ | 345 |
| 7 | Measurement Tone | 600 | 945 |
| 8 | Photon Depletion / Idling | 55 | **1000** |

(a) Z-Stabilizer Circuit

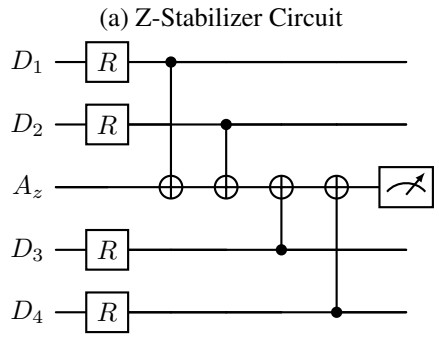

(b) X-Stabilizer Circuit

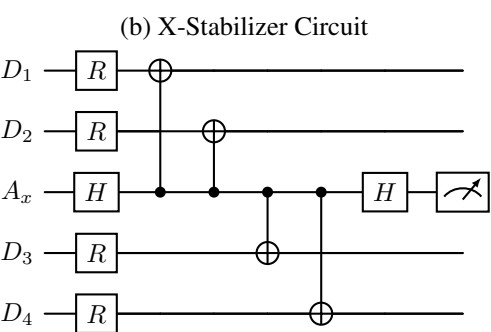

*Figure 5.* **Syndrome extraction circuits.** Standard circuit-level implementation of weight-4 stabilizers. **(a)** The Z-stabilizer measures parity $Z_1 Z_2 Z_3 Z_4$. **(b)** The X-stabilizer measures parity $X_1 X_2 X_3 X_4$, utilizing Hadamard gates on the ancilla to facilitate measurement in the X-basis. $D$ denotes data qubits, $A$ denotes stabilizers, and $R$ denotes the unexpected errors occur on data qubits. The circuit width is scaled to fit the column.

### A.3. Circuit-level Noise Modeling

Validating decoder performance prior to physical deployment relies on circuit-level noise simulations that inject Pauli errors after every gate operation. The choice of error model dictates the complexity of the decoding task. On one end of the spectrum lie physically derived models, such as the "Superconducting-Inspired" (SI1000) model (Gidney et al., 2021). These models incorporate detailed hardware asymmetries—including $T_1/T_2$ relaxation, leakage, and measurement bias—and typically reflect SOTA coherence limits with gate error rates in the range of $0.1\% \sim 0.3\%$.

On the other end is the standard depolarizing (SD) model (another error model introduced in (Gidney et al., 2021)), which applies isotropic noise parameterized by a single physical error probability $p$. While SI1000 offers fidelity to specific hardware implementations, this work primarily adopts the SD model configured at a near-threshold error rate of $p = 0.005$. This choice serves as a rigorous complexity stress test: the elevated error rate ($p = 0.5\%$, compared to $\sim 0.1\%$ in SI1000) generates significantly denser syndrome graphs. By validating real-time feasibility under this high-entropy, computationally intensive regime, we ensure that the decoder possesses sufficient headroom to handle the sparser, albeit more biased, errors characteristic of current superconducting processors.

### A.4. Real-time Constraints and Problem Formulation

**Hardware-imposed causality.** Practical QEC requires the decoder to operate strictly within the hardware clock cycle to prevent an irreversible backlog of error syndromes. As detailed in Table 3, the control timeline of superconducting processors is dominated by readout and reset operations, leaving a rigid latency budget of $t_{\text{cycle}} \approx 1\mu s$. Consequently, the decoding process must be strictly causal: at each timestep $t$, the decoder must infer corrections using only past information. To maintain fault tolerance under this constraint, we employ a sliding-window formulation where the temporal window depth $r$ scales with the spatial distance ($r = d$). This setting ensures that the logical error rate is suppressed exponentially in all spatiotemporal dimensions (isotropism), preventing "timelike" logical errors from accumulating across boundaries.

**The shift to per-shot latency.** This real-time operational regime necessitates a fundamental shift in evaluation metrics. Prior works often quantify decoder capability using **throughput** (amortized inference time per round). While high throughput is sufficient for passive memory experiments where corrections can be calculated in post-processing, it is physically invalid for active logical circuits where executing a gate requires an immediate update to the Pauli frame. The control system cannot tolerate the latency of a buffered sequence, even if the average speed is high. Therefore, in this work, we strictly evaluate feasibility using **per-shot latency**. We employ a continuous FIFO syndrome queue feeding the decoder, and measure the wall-clock time required to resolve a specific correction immediately at each clock cycle. This metric captures the true critical path of active error correction.

### A.5. Basic Concepts of FPGA

Here, we provide a brief overview of the fundamental building blocks of Field-Programmable Gate Arrays (FPGAs). Unlike CPUs or GPUs, which possess fixed datapaths (e.g., fixed number of ALUs or Tensor Cores), FPGAs consist of a "sea" of programmable logic resources that can be wired together to create custom digital circuits. The feasibility of a neural decoder

is determined by its consumption of three primary resource pools in FPGAs. As shown in the right panel of Fig. 2, they are Look-Up Tables (LUTs), Digital Signal Processors (DSPs), and Block RAM (BRAM).

**LUTs: the logic atoms.** LUTs are the finest-grained configurable units on an FPGA. A typical LUT (e.g., in Xilinx UltraScale architecture) is a small memory array that implements any Boolean function of 6 inputs. The key features of LUTs are summarized below.

- *Function:* While primarily used for control logic, LUTs can be chained together to perform arithmetic operations such as addition and multiplication.

- *Trade-off:* Implementing high-precision arithmetic (e.g., FP32 multiplication) using strictly LUTs is prohibitively expensive in terms of area. However, for ultra-low precision (e.g., INT4), LUTs become highly efficient.

- *Relevance:* In this work, we leverage the abundance of LUTs ($\sim 10^6$ per device) to implement massive parallel arrays of 4-bit multipliers, bypassing the bottleneck of specialized DSP blocks.

**DSPs: the arithmetic engines.** DSPs are specialized, hard-wired silicon blocks designed for high-speed, high-precision arithmetic (e.g., $27 \times 18$-bit multiplication + 48-bit accumulation). The key features of DSPs are summarized below.

- *Function:* They are the FPGA equivalent of a CPU's FPU or a GPU's Tensor Core. Standard deep learning accelerators typically map one MAC operation to one DSP slice.

- *Scarcity:* DSPs are the scarcest resource on an FPGA (typically only $1{,}000 \sim 3{,}000$ per device). Reliance on DSPs imposes a hard ceiling on the maximum parallelism, and thus the minimum latency, of the decoder.

- *Relevance:* Our INT4 quantization strategy proposed in Sec. 3.2 is explicitly designed to decouple the decoder from this scarce resource, allowing us to scale the architecture beyond the limits of available DSPs.

**BRAM: the on-chip storage.** BRAMs are discrete blocks of static RAM (typically 36 KB each) distributed throughout the FPGA fabric. The key features of BRAM are summarized below.

- *Function:* They serve as high-speed, low-latency local caches. Unlike off-chip memory (DRAM/HBM), which incurs hundreds of cycles of latency, BRAMs can provide data in a single clock cycle.

- *Relevance:* To achieve microsecond-scale decoding, the entire neural network weights must fit within the aggregate BRAM capacity ($\approx 30 \sim 100$ MB on high-end devices). Our compression pipeline reduces the model size to $< 5$ MB, ensuring comfortable residency in BRAM and eliminating the "memory wall."

**Key resource constraints.** Modern FPGA architectures (e.g., Xilinx UltraScale+) utilize BRAM for storage, DSPs for high-precision arithmetic, and LUTs for general logic. As indicated in Sec. 4.3, we identify **LUT availability as the dominant system constraint** for real-time neural decoding. Here, we expand this conclusion from the perspective of fundamental resource economics of FPGAs.

- *Shift from DSP to Logic:* While standard INT8/FP16 operations heavily rely on scarce DSP slices (typically $\sim 10^3$ per device), our INT4 arithmetic allows the synthesis toolchain to bypass DSPs entirely, implementing multiplication via LUTs, which are orders of magnitude more abundant ($\sim 10^6$). This transitions the system from a *DSP-bound* to a *Logic-bound* regime, unlocking massive parallelism that would otherwise be throttled by DSP scarcity.

- *Memory Sufficiency:* Regarding storage, the aggressive move to INT4 reduces the model size of $N$ million parameters to approximately $N/2$ MB. This footprint is negligible compared to the on-chip BRAM capacity of high-end FPGAs ($> 50$ MB), ensuring that the entire decoder fits on-chip to eliminate the latency penalty of DRAM access.

Consequently, we quantify hardware efficiency primarily through LUT utilization ($R_{\text{LUT}}$), treating DSPs only as auxiliary accelerators where applicable.

**Multiply-Accumulate (MAC).** The computational workload of a neural network is quantified in Multiply-Accumulate (MAC) operations, i.e.,

$$a \leftarrow a + (w \cdot x) \tag{10}$$

where $w$ is a weight parameter, $x$ is an input activation, and $a$ is the accumulator. In the context of hardware synthesis, a "MAC" is not a fixed physical component but a logical operation that must be mapped to physical resources:

- *DSP-based MAC:* For INT8 or FP32, one logical MAC typically consumes 1 physical DSP slice.

- *LUT-based MAC:* For INT4, one logical MAC is synthesized into a cluster of LUTs (approx. $1.5 \sim 20$ LUTs depending on programmability).

This flexible mapping capability allows us to trade abundant LUTs for scarce DSPs, optimizing the hardware specifically for the quantized distribution of the decoder.

## B. Related Work

**Quantum error-correcting codes**. Much work on fault-tolerant QEC decoding targets stabilizer codes, whose structure determines the syndrome representation and the algorithmic paradigms that are most commonly used for decoding. Surface codes (Fowler et al., 2012) and related topological families (e.g., color codes (Bombin & Martin-Delgado, 2006)) employ geometrically local stabilizer checks on a 2D lattice, yielding syndrome/detection-event patterns with predominantly local correlations; decoding thus operates on structured 2D or 3D spacetime graphs amenable to matching and local graph algorithms (deMarti iOlius et al., 2024). In contrast, quantum LDPC (QLDPC) codes (Panteleev & Kalachev, 2021b; Bravyi et al., 2024) achieve higher encoding rates through sparse but non-local parity checks, yielding irregular Tanner graphs where iterative message-passing methods such as belief propagation are commonly used, albeit with well-known challenges from short cycles and degeneracy (Panteleev & Kalachev, 2021a). Under circuit-level noise, repeated syndrome extraction introduces a temporal dimension, transforming decoding into a spatiotemporal inference problem across code families. The decoding methods surveyed in the following sections are therefore organized around these distinct graph structures and their associated spatiotemporal correlations.

**Traditional QEC decoders**. Traditional QEC decoders for stabilizer codes can be broadly grouped by the graph structures induced by the code family and noise model. In the surface-code setting, a standard approach maps detection events from the spacetime syndrome volume to a graph and solves an optimization problem to select a correction consistent with the observed syndrome. The most widely used baseline is minimum-weight perfect matching (MWPM), which reduces decoding under common assumptions to a minimum-weight matching problem solvable via blossom-type algorithms and efficient implementations (Edmonds, 1965; Fowler et al., 2012). Recent implementations exploit sparsity and streaming-style dataflow to reduce overhead at scale (Higgott & Gidney, 2025). Union-Find decoders provide an alternative on the same lattice structure, replacing global optimization with local cluster growth and disjoint-set merging, achieving almost-linear-time complexity in practice (Delfosse & Nickerson, 2021). Comprehensive surveys further discuss MWPM, Union-Find, and related local graph algorithms in the surface-code decoding literature (deMarti iOlius et al., 2024).

For codes defined on irregular Tanner (factor) graphs—including many quantum LDPC (qLDPC) families—iterative message passing such as belief propagation (BP) provides a natural decoding framework with strong classical LDPC precedent. In the quantum setting, however, short cycles and code degeneracy can hinder naive BP convergence and accuracy (Poulin & Chung, 2008). Augmented variants that combine BP with post-processing, such as ordered-statistics decoding (BP+OSD), improve robustness at additional computational cost and have become a common baseline in qLDPC studies (Roffe et al., 2020).

**Neural QEC decoders**. For topological codes whose syndromes admit regular spatiotemporal representations, a range of neural architectures has been explored on grid/volume inputs. Early work applied multilayer perceptrons (MLPs) to flattened syndrome vectors (Krastanov & Jiang, 2017; Torlai & Melko, 2017). Convolutional neural networks (CNNs) leverage locality and translation invariance by treating syndrome histories as 2D or 3D volumes (Varsamopoulos et al., 2019). To capture temporal correlations across measurement rounds, sequence models including recurrent networks and temporal convolutional networks (TCNs) have been adopted (Baireuther et al., 2018; Varbanov et al., 2025). Transformer-based architectures further enable long-range interactions and can incorporate soft measurement information when available (Bausch et al., 2024; Senior et al., 2025).

Situated between the strict locality of CNNs and the global receptive fields of Transformers, graph neural networks (GNNs) offer a flexible paradigm that can operate on either the detector graph (Lange et al., 2025; Maan & Paler, 2025) or the Tanner graph (Liu & Poulin, 2019). For topological codes like the surface code (our main focus in this paper), the detector graph forms a regular planar lattice. On such structures, a locally constrained GNN mathematically reduces to a standard CNN, while a globally connected GNN functions similarly to a Transformer but often with less optimized computational overhead. Due to this structural equivalence, standard CNNs or Transformers are typically preferred over explicit GNN formulations for surface codes. However, this dynamic changes fundamentally for non-local codes, such as quantum LDPC codes. Because these codes lack a grid-like planar structure, standard spatial convolutions become inapplicable. Consequently, GNNs emerge as an essential architectural necessity to capture their irregular connectivity, regardless of whether they are applied to the detector graph or the Tanner graph.

Representing a fundamentally distinct paradigm from the aforementioned detector-graph models, neural decoders operating directly on the Tanner graph possess the universal capability to adapt to almost any quantum code. A foundational example is neural belief propagation, which unrolls standard BP iterations into a trainable network and learns message-update rules end-to-end (Liu & Poulin, 2019). Other directions include hypergraph formulations for multi-body stabilizer constraints (Bhave et al., 2025), generative modeling of syndrome–logical mappings (Cao et al., 2023), and uncertainty-aware neural decoding that outputs confidence estimates (Mi & Mueller, 2025).

**Hardware implementations and real-time constraints**. Real-time fault-tolerant QEC imposes tight latency budgets on decoding, motivating dedicated hardware implementations—often on FPGAs—to sustain streaming syndrome processing within control-cycle constraints (Battistel et al., 2023). For classical decoders, several hardware-oriented designs have been demonstrated. Helios implements Union-Find decoding on FPGA with a scalable, distributed architecture tailored to surface-code lattices (Liyanage et al., 2024). For matching-based decoding, Micro Blossom accelerates MWPM-style computation via specialized hardware dataflow and incremental processing (Wu et al., 2025). Clustering-style decoders have also been mapped to hardware, including collision-clustering and related local-growth approaches that emphasize regular control flow and predictable latency (Barber et al., 2025). For irregular Tanner-graph codes, hardware-optimized BP variants such as Relay-BP restructure BP-style updates to better match implementation constraints (Maurer et al., 2025).

Beyond classical algorithms, a smaller body of work explores hardware-aware neural decoder design and deployment. Overwater et al. examine design-space trade-offs for feedforward decoders under resource and latency constraints at small code distances (Overwater et al., 2022).

Our work differs from prior studies on QEC decoding in its scope and emphasis. In particular, we focus on a systematic analysis under explicit accuracy–latency constraints, resulting in limited overlap with existing approaches.

## C. Model Implementation Details

In this section, we provide the implementation details of the five neural decoders omitted in the main text.

### C.1. Model Configurations

**MLP.** The MLP decoder takes the 1D detector-event vector $s \in \{0, 1\}^{n_{\text{det}}}$ as input, where $n_{\text{det}}$ depends on the code distance $d$ and number of measurement rounds $r$. An input projection maps $s$ to hidden_dim via Linear $\rightarrow$ LayerNorm $\rightarrow$ GELU $\rightarrow$ Dropout(0.1). The network then applies six residual blocks, each consisting of Linear $\rightarrow$ LayerNorm $\rightarrow$ GELU $\rightarrow$ Dropout(0.1) $\rightarrow$ Linear $\rightarrow$ LayerNorm, with an identity skip connection added across the block. A final linear layer produces a single output logit. We define two configurations: **Small** (hidden_dim = 512) and **Large** (hidden_dim = 1024). Table 4 lists the per-layer input and output dimensions.

*Table 4.* MLP per-layer architecture. Small: hidden_dim = 512; Large: hidden_dim = 1024.

| Stage | Layer | Input → Output | Small | Large |
|---|---|---|---|---|
| Input Projection | Linear | $n_{\text{det}}$ → hidden_dim | $n_{\text{det}}$→512 | $n_{\text{det}}$→1024 |
| | LayerNorm | hidden_dim | 512 | 1024 |
| | GELU | – | – | – |
| | Dropout(0.1) | – | – | – |
| Residual Block × 6 (per block) | Linear | hidden_dim → hidden_dim | 512→512 | 1024→1024 |
| | LayerNorm | hidden_dim | 512 | 1024 |
| | GELU | – | – | – |
| | Dropout(0.1) | – | – | – |
| | Linear | hidden_dim → hidden_dim | 512→512 | 1024→1024 |
| | LayerNorm | hidden_dim | 512 | 1024 |
| Output | Linear | hidden_dim → 1 | 512→1 | 1024→1 |

**3D-CNN.** The 3D-CNN decoder takes the 3D spatiotemporal tensor $(B, 2, T, H, W)$ produced by StimTo3DMapper as input, where $T$, $H$, and $W$ depend on the code distance $d$ and rounds $r$. A stem layer applies Conv3d($2 \to c_1$, kernel size 3, padding 1) followed by BatchNorm3d and ReLU. Three ResidualBlock3D modules follow with dilations 1, 2, 3 and channel configurations $c_1 \to c_1$, $c_1 \to c_2$, $c_2 \to c_2$ respectively; all convolutions use stride 1, so the spatial dimensions $(T, H, W)$ are preserved throughout. Each ResidualBlock3D applies

$$\text{Conv3d} \to \text{BatchNorm3d} \to \text{ReLU} \to \text{Dropout3d(0.1)} \to \text{Conv3d} \to \text{BatchNorm3d}$$

on the main branch, with a skip connection that uses identity when channels are unchanged or a $1\times1\times1$ Conv3d projection + BatchNorm3d otherwise, followed by ReLU. A $1\times1\times1$ bottleneck Conv3d reduces channels from $c_2$ to $c_{\text{bottle}}$, followed by BatchNorm3d and ReLU. The classifier flattens the feature volume and applies

$$\text{Linear} \to \text{BatchNorm1d} \to \text{ReLU} \to \text{Dropout(0.5)} \to \text{Linear}$$

to produce a single logit, where $c_{\text{fc}} = c_{\text{bottle}} \cdot T \cdot H \cdot W / 4$.

We define two configurations: **Small** ($c_1$=16, $c_2$=32, $c_{\text{bottle}}$=4) and **Large** ($c_1$=32, $c_2$=64, $c_{\text{bottle}}$=8). Table 5 lists the per-layer input and output dimensions.

**TCN.** The TCN decoder takes the 1D detector-event vector $\boldsymbol{s} \in \{0, 1\}^{n_{\text{det}}}$ as input, where $n_{\text{det}}$ depends on the code distance $d$ and number of measurement rounds $r$. A ConstantEmbedding2D front-end maps each syndrome bit to a fixed $c_1$-dimensional feature vector at its physical lattice coordinate, producing a spatiotemporal tensor $(B, r, c_1, H, W)$ where $H \times W$ is the 2D stabilizer grid. The employed mapping is randomly initialized and non-trainable. The spatial encoder then begins with a ResConv2D block whose main branch applies

$$\text{Conv2d} \to \text{BatchNorm2d} \to \text{Dropout2d(0.1)},$$

preserving $c_1$, followed by a final ReLU after the residual add. Three additional Conv2d layers, each followed by BatchNorm2d and ReLU, then form the rest of the spatial encoder with channel configurations $c_1 \to c_1$, $c_1 \to c_2$, and $c_2 \to c_2$. All 2D convolutions use kernel size 3 with padding 1 and stride 1, so that the spatial dimensions $(H, W)$ are preserved throughout.

The resulting $(B, r, c_2, H, W)$ tensor is permuted and flattened into a token sequence $(B, S, c_2)$ with $S = r \cdot H \cdot W$, then transposed to $(B, c_2, S)$ and passed through two temporal convolutional blocks, each applying

$$\text{Conv1d} \to \text{BatchNorm1d} \to \text{ReLU}$$

with kernel size 3 and padding 1, preserving the channel count ($c_2 \to c_2$).

The classifier applies a per-position linear projection ($c_2 \to c_2$) followed by ReLU, then mean pooling over the sequence dimension, LayerNorm($c_2$), and a final linear layer to a single logit.

As stated in the main text, we consider two configurations: **Small** ($c_1$=32, $c_2$=64) and **Large** ($c_1$=64, $c_2$=128). Table 6 lists the per-layer input and output dimensions.

*Table 5.* 3D-CNN per-layer architecture. Small: $c_1{=}16$, $c_2{=}32$, $c_{\text{bottle}}{=}4$; Large: $c_1{=}32$, $c_2{=}64$, $c_{\text{bottle}}{=}8$. Spatial dimensions $(T, H, W)$ depend on the code distance and are preserved throughout. $c_{\text{fc}} = c_{\text{bottle}} \cdot T \cdot H \cdot W\,/\,4$.

| Stage | Layer | Input → Output | Small | Large |
|---|---|---|---|---|
| Stem | Conv3d($k{=}3, p{=}1$) | $2 \to c_1$ | $2{\to}16$ | $2{\to}32$ |
| | BatchNorm3d | $c_1$ | 16 | 32 |
| | ReLU | – | – | – |
| ResBlock 1 (dilation 1) | Conv3d($k{=}3$, dil=1, $p{=}1$) | $c_1 \to c_1$ | $16{\to}16$ | $32{\to}32$ |
| | BatchNorm3d | $c_1$ | 16 | 32 |
| | ReLU | – | – | – |
| | Dropout3d(0.1) | – | – | – |
| | Conv3d($k{=}3$, dil=1, $p{=}1$) | $c_1 \to c_1$ | $16{\to}16$ | $32{\to}32$ |
| | BatchNorm3d | $c_1$ | 16 | 32 |
| | + Identity, ReLU | – | – | – |
| ResBlock 2 (dilation 2) | Conv3d($k{=}3$, dil=2, $p{=}2$) | $c_1 \to c_2$ | $16{\to}32$ | $32{\to}64$ |
| | BatchNorm3d | $c_2$ | 32 | 64 |
| | ReLU | – | – | – |
| | Dropout3d(0.1) | – | – | – |
| | Conv3d($k{=}3$, dil=2, $p{=}2$) | $c_2 \to c_2$ | $32{\to}32$ | $64{\to}64$ |
| | BatchNorm3d | $c_2$ | 32 | 64 |
| | Shortcut: Conv3d($1{\times}1{\times}1$) + BN | $c_1 \to c_2$ | $16{\to}32$ | $32{\to}64$ |
| | + Shortcut, ReLU | – | – | – |
| ResBlock 3 (dilation 3) | Conv3d($k{=}3$, dil=3, $p{=}3$) | $c_2 \to c_2$ | $32{\to}32$ | $64{\to}64$ |
| | BatchNorm3d | $c_2$ | 32 | 64 |
| | ReLU | – | – | – |
| | Dropout3d(0.1) | – | – | – |
| | Conv3d($k{=}3$, dil=3, $p{=}3$) | $c_2 \to c_2$ | $32{\to}32$ | $64{\to}64$ |
| | BatchNorm3d | $c_2$ | 32 | 64 |
| | + Identity, ReLU | – | – | – |
| Bottleneck | Conv3d($k{=}1$) | $c_2 \to c_{\text{bottle}}$ | $32{\to}4$ | $64{\to}8$ |
| | BatchNorm3d | $c_{\text{bottle}}$ | 4 | 8 |
| | ReLU | – | – | – |
| Classifier | Flatten | $c_{\text{bottle}}{\cdot}T{\cdot}H{\cdot}W$ | $4{\cdot}T{\cdot}H{\cdot}W$ | $8{\cdot}T{\cdot}H{\cdot}W$ |
| | Linear | $c_{\text{bottle}}{\cdot}THW \to c_{\text{fc}}$ | $4THW{\to}THW$ | $8THW{\to}2THW$ |
| | BatchNorm1d | $c_{\text{fc}}$ | $THW$ | $2THW$ |
| | ReLU | – | – | – |
| | Dropout(0.5) | – | – | – |
| | Linear | $c_{\text{fc}} \to 1$ | $THW{\to}1$ | $2THW{\to}1$ |

**Transformer.** The Transformer decoder employs a SpatioTemporalEncoder (STE) front-end that takes the 3D spatiotemporal tensor $(B, 2, T, H, W)$ produced by StimTo3DMapper as input. The STE applies a Conv3d stem ($2 \to c_1$, kernel size 3, padding 1) followed by BatchNorm3d and ReLU. Three ResidualBlock3D modules follow, using the same internal structure as in 3D-CNN (Conv3d → BN → ReLU → Dropout3d(0.1) → Conv3d → BN → skip → ReLU): $c_1{\to}c_2$ with stride 1 and dilation 1; $c_2{\to}c_2$ with stride $(2, 1, 1)$ and dilation $(1, 2, 2)$, performing temporal downsampling to $T' \approx T/2$; and $c_2{\to}c_3$ with stride 1 and dilation $(1, 3, 3)$. A $1{\times}1{\times}1$ Conv3d projects from $c_3$ to embed_dim; the output is permuted to $(B, T', H, W, \text{embed\_dim})$. Before flattening, a 3D sinusoidal positional embedding is added along the $T'$, $H$, and $W$ axes independently, then flattened to $(B, S, \text{embed\_dim})$ with $S = T' \cdot H \cdot W$.

The token sequence is processed by a Pre-LN Transformer encoder with 3 layers (`norm_first=True`). Each layer applies

LayerNorm → MultiheadAttention → Dropout(0.1) → residual add → LayerNorm → Linear(embed_dim → ffn_dim),

*Table 6.* TCN per-layer architecture. Small: $c_1$=32, $c_2$=64. Large: $c_1$=64, $c_2$=128. Spatial dimensions $(r, H, W)$ depend on the code distance; $S = r \cdot H \cdot W$.

| Stage | Layer | Input $\rightarrow$ Output | Small | Large |
|---|---|---|---|---|
| Embedding | ConstantEmbedding2D | $s \rightarrow (B, r, c_1, H, W)$ | $c_1$=32 | $c_1$=64 |
| ResBlock | Conv2d($k$=3, $p$=1) | $c_1 \rightarrow c_1$ | 32$\rightarrow$32 | 64$\rightarrow$64 |
| | BatchNorm2d | $c_1$ | 32 | 64 |
| | Dropout2d(0.1) | – | – | – |
| | Residual add + ReLU | – | – | – |
| Conv2D $\times$3 | Conv2d($k$=3, $p$=1) | $c_1 \rightarrow c_1$ | 32$\rightarrow$32 | 64$\rightarrow$64 |
| | Conv2d($k$=3, $p$=1) | $c_1 \rightarrow c_2$ | 32$\rightarrow$64 | 64$\rightarrow$128 |
| | Conv2d($k$=3, $p$=1) | $c_2 \rightarrow c_2$ | 64$\rightarrow$64 | 128$\rightarrow$128 |
| Flatten | Permute | $(B, r, c_2, H, W) \rightarrow (B, r, H, W, c_2)$ | – | – |
| | Reshape | $\rightarrow (B, S, c_2)$ | $\rightarrow (B, S, 64)$ | $\rightarrow (B, S, 128)$ |
| | Transpose | $\rightarrow (B, c_2, S)$ | $\rightarrow (B, 64, S)$ | $\rightarrow (B, 128, S)$ |
| TCN | Conv1d($k$=3, $p$=1) | $c_2 \rightarrow c_2$ | 64$\rightarrow$64 | 128$\rightarrow$128 |
| | Conv1d($k$=3, $p$=1) | $c_2 \rightarrow c_2$ | 64$\rightarrow$64 | 128$\rightarrow$128 |
| Classifier | Linear (per position) | $c_2 \rightarrow c_2$ | 64$\rightarrow$64 | 128$\rightarrow$128 |
| | ReLU | – | – | – |
| | MeanPool(dim=seq) | $(B, S, c_2) \rightarrow (B, c_2)$ | $\rightarrow (B, 64)$ | $\rightarrow (B, 128)$ |
| | LayerNorm | $c_2$ | 64 | 128 |
| | Linear | $c_2 \rightarrow 1$ | 64$\rightarrow$1 | 128$\rightarrow$1 |

then

$$\text{ReLU} \rightarrow \text{Dropout}(0.1) \rightarrow \text{Linear}(\text{ffn\_dim} \rightarrow \text{embed\_dim}) \rightarrow \text{Dropout}(0.1) \rightarrow \text{residual add},$$

where $\text{ffn\_dim} = 4 \cdot \text{embed\_dim}$.

The classifier applies mean pooling over the sequence dimension, followed by LayerNorm(embed_dim) and a linear layer to a single logit. We define two configurations: **Small**, i.e., STE: $c_1$=8, $c_2$=16, $c_3$=32; embed_dim=64, num_heads=4, ffn_dim=256) and **Large**, i.e., STE: $c_1$=16, $c_2$=32, $c_3$=64; embed_dim=128, num_heads=8, ffn_dim=512. Table 7 lists the per-layer input and output dimensions.

*Table 7.* Transformer per-layer architecture. Small: STE $c_1$=8, $c_2$=16, $c_3$=32; embed_dim=64, num_heads=4, ffn_dim=256. Large: STE $c_1$=16, $c_2$=32, $c_3$=64; embed_dim=128, num_heads=8, ffn_dim=512. Spatial dimensions $(T, H, W)$ depend on the code distance; $S = T' \cdot H \cdot W$.

| Stage | Layer | Input → Output | Small | Large |
|---|---|---|---|---|
| STE: Stem | Conv3d($k$=3, $p$=1) | $2 \rightarrow c_1$ | 2→8 | 2→16 |
| | BatchNorm3d | $c_1$ | 8 | 16 |
| | ReLU | – | – | – |
| STE: Backbone (ResBlock3D ×3) | ResBlock3D($s$=1, dil=1) | $c_1 \rightarrow c_2$ | 8→16 | 16→32 |
| | ResBlock3D($s$=(2, 1, 1), dil=(1, 2, 2)) | $c_2 \rightarrow c_2$  [$T \rightarrow T'$] | 16→16 | 32→32 |
| | ResBlock3D($s$=1, dil=(1, 3, 3)) | $c_2 \rightarrow c_3$ | 16→32 | 32→64 |
| STE: Projection | Conv3d($k$=1) | $c_3 \rightarrow$ embed_dim | 32→64 | 64→128 |
| | + 3D Sinusoidal PE, Permute + Flatten | $\rightarrow (B, S, \text{embed\_dim})$ | $\rightarrow (B, S, 64)$ | $\rightarrow (B, S, 128)$ |
| Transformer Encoder × 3 (per layer) | *Self-Attention Block* | | | |
| | LayerNorm | embed_dim | 64 | 128 |
| | MultiheadAttention | (embed_dim, num_heads) | (64, 4) | (128, 8) |
| | Dropout(0.1) | – | – | – |
| | + Residual | – | – | – |
| | *Feed-Forward Block* | | | |
| | LayerNorm | embed_dim | 64 | 128 |
| | Linear | embed_dim → ffn_dim | 64→256 | 128→512 |
| | ReLU + Dropout(0.1) | – | – | – |
| | Linear | ffn_dim → embed_dim | 256→64 | 512→128 |
| | Dropout(0.1) + Residual | – | – | – |
| Classifier | MeanPool(dim=1) | $(B, S, \text{embed\_dim}) \rightarrow (B, \text{embed\_dim})$ | →(B, 64) | →(B, 128) |
| | LayerNorm | embed_dim | 64 | 128 |
| | Linear | embed_dim → 1 | 64→1 | 128→1 |

**GNN (Neural BP).** The GNN decoder operates on a bipartite graph constructed from the detector error model (DEM) via StimToGraphMapper, with check nodes (detectors) and variable nodes (physical errors) connected by edges derived from the DEM. Check node features are formed by centering the syndrome bit $s \in \{0, 1\}$ as $1 - 2s$ and projecting via Linear($1 \to$ hidden_dim); variable node features are initialized from log-likelihood ratios and projected via Linear($1 \to$ hidden_dim).

Message passing is performed by a single NeuralBPLayer applied iteratively for $N$ iterations with shared weights, where $N \in \{d, 2d\}$. Each iteration consists of two phases. In the V→C phase, variable node states are aggregated (sum) at each check node and concatenated with the initial check features to form a hidden_dim×2 vector; a V2C MLP (Linear $\to$ LayerNorm $\to$ ScaledTanh $\to$ Dropout(0.1) $\to$ Linear) processes this vector, and a GRUCell updates the check hidden state. In the C→V phase, updated check node states are aggregated (sum) at each variable node; a C2V MLP (Linear $\to$ LayerNorm $\to$ ScaledTanh $\to$ Dropout(0.1) $\to$ Linear) processes the aggregated message, and a GRUCell updates the variable hidden state. ScaledTanh is a learnable-scale tanh activation with a clamped scale parameter in $[0.1, 5.0]$.

A per-variable readout head applies Linear $\to$ GELU $\to$ Dropout(0.1) $\to$ Linear to produce one logit per variable node. Physical error predictions are obtained by thresholding, and the logical flip is computed as $\hat{y} = (\hat{e} \cdot L) \bmod 2$ using the logical operator matrix $L$. We define two configurations: **Small** (hidden_dim=32) and **Large** (hidden_dim=64). Table 8 lists the per-layer input and output dimensions.

*Table 8.* GNN (Neural BP) per-layer architecture. Small: hidden_dim=32; Large: hidden_dim=64. Message passing iterations $N \in \{d, 2d\}$.

| Stage | Layer | Input → Output | Small | Large |
|---|---|---|---|---|
| Feature Encoding | Check: Linear | $1 \to$ hidden_dim | $1{\to}32$ | $1{\to}64$ |
| | Variable: Linear | $1 \to$ hidden_dim | $1{\to}32$ | $1{\to}64$ |
| NeuralBP $\times N$: V→C phase | Sum Aggregation | hidden_dim | 32 | 64 |
| | Concat [aggr, $h_c^{(0)}$] | hidden_dim×2 | 64 | 128 |
| | Linear | hidden_dim×2 $\to$ hidden_dim | $64{\to}32$ | $128{\to}64$ |
| | LayerNorm | hidden_dim | 32 | 64 |
| | ScaledTanh + Dropout(0.1) | – | – | – |
| | Linear | hidden_dim $\to$ hidden_dim | $32{\to}32$ | $64{\to}64$ |
| | Check GRUCell | hidden_dim | 32 | 64 |
| NeuralBP $\times N$: C→V phase | Sum Aggregation | hidden_dim | 32 | 64 |
| | Linear | hidden_dim $\to$ hidden_dim | $32{\to}32$ | $64{\to}64$ |
| | LayerNorm | hidden_dim | 32 | 64 |
| | ScaledTanh + Dropout(0.1) | – | – | – |
| | Linear | hidden_dim $\to$ hidden_dim | $32{\to}32$ | $64{\to}64$ |
| | Variable GRUCell | hidden_dim | 32 | 64 |
| Readout | Linear | hidden_dim $\to$ hidden_dim | $32{\to}32$ | $64{\to}64$ |
| | GELU | – | – | – |
| | Dropout(0.1) | – | – | – |
| | Linear | hidden_dim $\to$ 1 | $32{\to}1$ | $64{\to}1$ |

## C.2. FP32 Parameter Statistics

Table 9 reports the full-precision (FP32) parameter counts for the three quantization-compatible architectures (3D-CNN, TCN, Transformer) across code distances $d \in \{3, 5, 7, 9\}$ and two model scales. Parameter counts are obtained by instantiating the model in PyTorch and calling `sum(p.numel() for p in model.parameters())`. These FP32 baselines serve as the starting point for the compression pipeline described in Sec. C.3.

**Key observations.** **3D-CNN** exhibits cubic parameter growth with code distance: from 146K at $d$=3 to 32.5M at $d$=9 (a $223\times$ increase), primarily due to the dense classifier head that scales with the 3D syndrome volume $(T, H, W)$. In contrast, **TCN** and **Transformer** maintain constant parameter counts across code distances within the same scale (TCN-Small: 103K for all $d$; Transformer-Small: 220K for all $d$), as their temporal processing modules operate on sequence representations

*Table 9.* **FP32 model parameter statistics.** Parameter counts are measured via PyTorch for the three architectures evaluated under quantization. Only 3D-CNN, TCN, and Transformer are included, as MLP and GNN are not quantized in our experiments.

| Model | $d$ | Scale | Parameters |
|---|---|---|---|
| 3D-CNN | 3 | Small | 145,801 |
| | 5 | Small | 487,497 |
| | 7 | Large | 8,844,305 |
| | 9 | Large | 32,463,505 |
| TCN | 3 | Small | 103,297 |
| | 5 | Small | 103,297 |
| | 7 | Large | 411,393 |
| | 9 | Large | 411,393 |
| Transformer | 3 | Small | 219,585 |
| | 5 | Small | 219,585 |
| | 7 | Large | 871,297 |
| | 9 | Large | 871,297 |

with fixed hidden dimensions independent of the input syndrome volume size. This architectural difference directly impacts storage requirements and computational complexity, as reflected in the effective MACs and clock-cycle estimates in Table 2.

### C.3. Quantization and Pruning

We implement model compression through quantization and pruning using the Brevitas library (Pappalardo et al., 2025), which provides hardware-aware quantization capabilities for PyTorch models. Quantization is applied to three architectures, i.e., 3D-CNN, TCN, and Transformer, that share convolutional and attention-based components amenable to low-bit representations.

**Quantization scheme.** We adopt symmetric uniform quantization for both weights and activations. Weights use **per-channel** quantization, where each output channel has an independent scale factor, enabling finer granularity and better preservation of weight distributions across channels. Activations use **per-tensor** quantization with a single scale factor shared across all elements. We evaluate two bit-width configurations: **W4A4** (4-bit weights and 4-bit activations) and **W8A8** (8-bit weights and 8-bit activations).

**Layer mapping.** Table 10 summarizes how each layer type is handled in FP32, PTQ, and QAT modes. Convolutional layers (Conv3d, Conv2d, Conv1d) and fully-connected layers (Linear) are replaced with their Brevitas counterparts (QuantConv3d, QuantConv2d, QuantConv1d, QuantLinear), which simulate quantization during forward passes.

For the Transformer-based model, the native `nn.MultiheadAttention` is replaced with `qnn.QuantMultiheadAttention`, which quantizes the Q/K/V/O projection weights while keeping the softmax operation in FP32. Normalization layers (BatchNorm, LayerNorm), activation functions (ReLU), and Dropout remain in FP32 throughout. However, BatchNorm receives different treatment in PTQ versus QAT. Specifically, in PTQ, BatchNorm parameters are algebraically folded into the preceding Conv or Linear layer to eliminate the normalization operation at inference; in QAT, BatchNorm remains as a separate FP32 layer during training. Residual additions in ResidualBlock3D use `qnn.QuantEltwiseAdd` to maintain quantized representations across skip connections.

**Weight mapping for pretrained models.** When initializing quantized models from pretrained FP32 weights, the state dictionary keys must match between the source and target models. For Conv3d, Conv2d, Conv1d, and Linear layers, Brevitas quantized layers use identical parameter naming conventions as their native PyTorch counterparts (e.g., `weight`, `bias`), so no key transformation is required—the FP32 weights can be loaded directly. LayerNorm, BatchNorm, ReLU, and Dropout layers remain as native PyTorch modules in both FP32 and quantized models, so their keys are inherently compatible.

The exception is `nn.MultiheadAttention`, which uses a different internal structure than `qnn.QuantMultiheadAttention`. The native PyTorch implementation stores the Q/K/V projections as a single fused weight tensor `in_proj_weight`, whereas Brevitas separates them into `in_proj.weight`. Similarly,

*Table 10.* Layer mapping for FP32, PTQ, and QAT modes. "Calibration" indicates the layer is quantized using activation statistics collected from training data. "Training" indicates the layer participates in quantization-aware training with simulated quantization. "Folded" means parameters are merged into the preceding layer.

| Component | FP32 Layer | Brevitas Layer | PTQ | QAT |
|---|---|---|---|---|
| Conv3d | `nn.Conv3d` | `qnn.QuantConv3d` | Calibration | Training |
| Conv2d | `nn.Conv2d` | `qnn.QuantConv2d` | Calibration | Training |
| Conv1d | `nn.Conv1d` | `qnn.QuantConv1d` | Calibration | Training |
| Linear | `nn.Linear` | `qnn.QuantLinear` | Calibration | Training |
| MultiheadAttn | `nn.MHA` | `qnn.QuantMHA` | Calibration | Training |
| BatchNorm | `nn.BatchNorm*` | – | Folded | FP32 |
| LayerNorm | `nn.LayerNorm` | – | FP32 | FP32 |
| ReLU | `nn.ReLU` | – | FP32 | FP32 |
| Dropout | `nn.Dropout*` | – | FP32 | FP32 |
| Residual Add | `+` | `qnn.QuantEltwiseAdd` | Quantized | Quantized |

`in_proj_bias` maps to `in_proj.bias`, and the output projection keys `out_proj.weight`/`out_proj.bias` remain unchanged. This mapping is only relevant for Transformer; 3D-CNN and TCN do not use attention layers and require no key transformation.

**Post-training quantization (PTQ).** PTQ converts a pretrained FP32 model to a quantized model without retraining, making it computationally efficient but potentially less accurate than QAT. The PTQ pipeline consists of two steps:

1. *BatchNorm Folding*: For each Conv–BatchNorm or Linear–BatchNorm pair, we compute folded weights $W' = W \cdot \gamma/\sqrt{\sigma^2 + \epsilon}$ and biases $b' = \beta + (b - \mu) \cdot \gamma/\sqrt{\sigma^2 + \epsilon}$, where $\gamma, \beta, \mu, \sigma^2$ are the BatchNorm parameters. The BatchNorm layer is then replaced with an identity operation.

2. *Calibration*: A subset of training samples is passed through the network to collect activation statistics (min/max values per tensor). These statistics determine the quantization scale factors $s = (\max - \min)/(2^b - 1)$ for $b$-bit quantization.

After calibration, the model can be evaluated directly without further training.

**Remark**. Since batch normalization (BN) reduces to a fixed affine transformation at inference time, we perform compile-time batch normalization folding, absorbing the BN parameters $(\gamma, \beta, \mu, \sigma)$ into the preceding convolutional weights $\mathbf{W}$ and bias $\mathbf{b}$ to align the training graph with the inference graph:

$$\mathbf{W}'_{\text{fold}} = \frac{\gamma \cdot \mathbf{W}}{\sqrt{\sigma^2 + \epsilon}}, \quad \mathbf{b}'_{\text{fold}} = \beta + \frac{\gamma(\mathbf{b} - \mu)}{\sqrt{\sigma^2 + \epsilon}}. \tag{11}$$

This ensures that PTQ is applied to the true inference graph.

**Quantization-aware training (QAT).** QAT incorporates quantization simulation into the training loop, allowing the model to learn weight distributions that are robust to quantization errors. During the forward pass, weights and activations are quantized using the current scale factors; during backpropagation, the straight-through estimator (STE) passes gradients through the non-differentiable quantization function as if it were an identity. QAT can start from random initialization or fine-tune from pretrained FP32 weights. When loading FP32 weights, the state dictionary keys are mapped according to Table 10 (relevant only for Transformer's MultiheadAttention layers; 3D-CNN and TCN use identical keys). QAT typically achieves higher accuracy than PTQ because the model adapts its weights to compensate for quantization-induced information loss.

**Pruning.** We apply unstructured magnitude-based pruning using the $\ell_1$-norm criterion, which sets the smallest-magnitude weights to zero. We adopt **global** pruning, which ranks all weights across the entire model by their absolute values and prunes the smallest fraction regardless of layer membership. This approach allows more critical layers to retain more parameters while less sensitive layers absorb greater sparsity, achieving better accuracy than layer-wise pruning at the same

overall sparsity ratio. After applying the pruning mask, we fine-tune the sparse model to recover accuracy using label smoothing. Pruning is applied on top of QAT models, producing sparse quantized networks that combine both compression techniques.

## D. Experimental Details

### D.1. Neural Decoder Benchmark

**Synthetic datasets.** We generate training and test data using the Stim library (Gidney, 2021) with the rotated surface code in memory-Z mode (`surface_code:rotated_memory_z`). The circuit-level noise model applies depolarizing errors at four locations: after Clifford gates, after reset operations, before each measurement round (data qubit idling errors), and before measurement operations, all with physical error rate $p = 0.005$. The number of measurement rounds equals the code distance ($r = d$). For code distances $d \in \{3, 5\}$, we evaluate training set sizes of $\{10^5, 5 \times 10^5, 10^6, 5 \times 10^6\}$ samples; for $d \in \{7, 9\}$, we use $\{5 \times 10^5, 10^6, 5 \times 10^6, 2.5 \times 10^7\}$ samples. All configurations use a test set of $5 \times 10^4$ samples, and each experiment is repeated three times with different random initializations.

Table 11 summarizes the training hyperparameters. All models are trained with the AdamW optimizer and a cosine annealing learning rate schedule with linear warmup. Weight decay is set to $10^{-3}$ for 3D-CNN, TCN, and Transformer, and $10^{-4}$ for MLP and GNN. All models use dropout rate 0.1. For GNN, we evaluate both $N = d$ and $N = 2d$ message passing iterations.

*Table 11.* Training hyperparameters for the neural decoder benchmark.

| Hyperparameter | Value |
| --- | --- |
| Optimizer | AdamW |
| Learning Rate | $5 \times 10^{-4}$ |
| LR Schedule | Cosine annealing |
|    Warmup Epochs | 5 |
|    Decay End Epoch | 80 |
|    Minimum LR | $10^{-6}$ |
| Batch Size | 16,384 $\times$ 2 (effective 32,768) |
| Epochs | 100 (max 150) |
| Early Stopping Patience | 10 |
| Gradient Clipping | 1.0 |
| Loss Function | BCEWithLogitsLoss |
| Weight Decay | $10^{-3}$ (3D-CNN/TCN/Transformer), $10^{-4}$ (MLP/GNN) |
| Dropout | 0.1 |

**Sycamore datasets.** We validate our models on experimental data from the Google Sycamore processor, publicly available at `https://zenodo.org/records/6804040`. We select configurations where the number of measurement rounds equals the code distance, specifically $d=3, r=3$ and $d=5, r=5$. For $d=3$, the dataset provides four center qubit locations; we select center $(7, 5)$ which exhibits the lowest baseline logical error rate among the four options. Using the device-calibrated Stim circuit (`circuit_noisy.stim`) provided with the dataset, we sample $5 \times 10^6$ synthetic training examples that reflect the experimentally characterized noise profile. Training hyperparameters match those used for synthetic data. The dataset contains $5 \times 10^4$ experimental shots per configuration; we split these into $4.5 \times 10^4$ samples for fine-tuning and $5 \times 10^3$ samples for testing. Fine-tuning adapts models pretrained on synthetic data to the experimental noise distribution.

**Baseline decoder.** We use PyMatching as MWPM baseline. The decoder is configured with a detector error model (DEM) generated by Stim using `decompose_errors=True`, which decomposes hyperedges (errors affecting more than two detectors) into graphlike edges suitable for matching. This DEM captures correlation information from the circuit-level noise model, providing edge weights that reflect the true error structure rather than assuming independent noise. As a result, our MWPM baseline represents a strong classical reference that leverages the full noise model information available from the quantum circuit.

**Evaluation metric.** We report the logical error rate (LER), defined as the fraction of decoding trials in which the decoder incorrectly predicts the logical observable flip:

$$\text{LER} = \frac{1}{N} \sum_{i=1}^{N} \mathbf{1}[\hat{y}_i \neq y_i], \tag{12}$$

where $N$ is the number of test samples, $y_i \in \{0, 1\}$ is the ground-truth logical flip for sample $i$, and $\hat{y}_i$ is the decoder's prediction. Lower LER indicates better decoding performance.

### D.2. Quantization and Pruning

Table 12 summarizes the training hyperparameters for quantization and pruning experiments. Quantization experiments are conducted on 3D-CNN, TCN, and Transformer; MLP and GNN do not have quantized implementations. As described in Appendix C.3, quantization is applied to all Conv3d, Conv2d, Conv1d, Linear, and MultiheadAttention layers, while BatchNorm, LayerNorm, ReLU, and Dropout remain in FP32.

**PTQ implementation.** PTQ converts a pretrained FP32 model to a quantized model *without retraining*. We load FP32 weights into the quantized architecture, apply BatchNorm folding to algebraically merge normalization parameters into the preceding Conv/Linear layers, and then calibrate quantization scales by collecting activation statistics over 50 training batches. The calibrated model is evaluated directly without any gradient updates. We evaluate PTQ under both W8A8 and W4A4 configurations.

**QAT implementation.** Unlike PTQ, QAT *fine-tunes* the model with quantization simulation enabled. We initialize the quantized model from pretrained FP32 weights; Brevitas automatically initializes the new quantization scale parameters. Fine-tuning runs for up to 50 epochs with early stopping (patience 5) using a learning rate of $10^{-4}$—lower than FP32 training ($5 \times 10^{-4}$) to ensure stable convergence under quantization noise. BatchNorm layers remain in FP32 and are *not* folded during QAT, allowing them to adapt to the quantized weight distributions. All QAT experiments use the W4A4 configuration.

**Pruning.** Pruning is applied to trained QAT models with W4A4. We use **global** $\ell_1$-norm unstructured pruning: all weights across the entire model are ranked by absolute magnitude, and the smallest fraction is set to zero regardless of which layer they belong to. This global ranking allows more critical layers to retain more parameters while less sensitive layers absorb greater sparsity, achieving better accuracy than layer-wise pruning at the same overall sparsity. The output (classifier) layer is protected from pruning to preserve prediction capacity. We evaluate sparsity ratios of 70%, 80%, and 90%. After applying the pruning mask, we fine-tune the sparse model for up to 30 epochs with early stopping (patience 5) using a reduced learning rate of $5 \times 10^{-5}$. Upon completion, pruning masks are removed to permanently zero out the pruned weights, yielding a sparse state dictionary.

### D.3. Resource Estimation

**FPGA platform selection.** We select two devices from AMD Xilinx's Versal Premium series as target platforms: the VP1802 and VP1902. This choice is motivated by the following considerations. First, the two devices span a 2.5× range in logic capacity (VP1802: 3.36M LUTs; VP1902: 8.46M LUTs), covering deployment scenarios from mid-tier to high-end and enabling a clear delineation of feasibility boundaries under different resource budgets. Second, both devices belong to the Versal ACAP architecture (7nm process), a product line optimized by AMD for emulation and prototyping applications. Notably, despite its larger logic capacity, the VP1902 contains fewer DSP slices (6,864) than the VP1802 (14,352), a design trade-off indicating optimization for logic-heavy workloads rather than traditional DSP-intensive computation. This architectural emphasis aligns perfectly with our INT4 logic-bound paradigm. Additionally, the Versal family supports operating frequencies of 300–500 MHz, sufficient to meet the $< 1\,\mu$s real-time latency constraint.

**Hardware specifications.** Table 13 summarizes the key specifications of the two target devices. We focus on parameters directly relevant to the resource estimation model (Sec. 3.3): total LUT count, available LUT count after derating, and the derived maximum parallelism $P_{\max}$.

*Table 12.* Training hyperparameters for quantization and pruning experiments.

| Hyperparameter | PTQ | QAT | Pruning Fine-tune |
|---|---|---|---|
| Pretrained Init | FP32 weights | FP32 weights | QAT weights |
| Training Samples | – | $5 \times 10^6$ | $5 \times 10^6$ |
| Epochs | – | 50 | 30 |
| Early Stopping | – | patience 5 | patience 5 |
| Learning Rate | – | $10^{-4}$ | $5 \times 10^{-5}$ |
| LR Schedule | – | Cosine (min $10^{-6}$) | Cosine (min $10^{-6}$) |
| Weight Decay | – | $10^{-3}$ | $10^{-3}$ |
| Gradient Clipping | – | 1.0 | 1.0 |
| Batch Size | 1024 | 4096 | 4096 |
| Calibration Batches | 50 | – | – |
| BN Folding | Yes | No | No |
| Quant Config | W8A8, W4A4 | W4A4 | W4A4 |
| Sparsity Ratios | – | – | 70%, 80%, 90% |

*Table 13.* **Target FPGA specifications.** Available LUTs reflect a 50% derating factor. Maximum parallelism $P_{\mathrm{max}}$ is computed as $N_{\mathrm{LUT}}^{\mathrm{avail}}/\beta$ with $\beta = 20$ LUTs per INT4 PE.

| Specification | VP1802 | VP1902 |
|---|---|---|
| Architecture | Versal Premium (7nm) | Versal Premium (7nm) |
| System Logic Cells | 7.35M | 18.51M |
| Total LUTs | 3.36M | 8.46M |
| Available LUTs (50% derating) | 1.68M | 4.23M |
| DSP Slices | 14,352 | 6,864 |
| Max Parallelism ($P_{\mathrm{max}}$) | 84,022 PEs | 211,507 PEs |
| Target Frequency ($f_{\mathrm{clk}}$) | 300–400 MHz | 300–400 MHz |

**Resource derating methodology.** In our resource estimation, we apply a 50% derating factor to the total LUT count. That is, we assume only half of the physical LUT resources are available for theoretical feasibility analysis. This conservative assumption is grounded in three considerations:

*(1) Routing Congestion.* FPGA usability is constrained not only by logic resources but also by fixed routing channels. Engineering practice shows that when LUT utilization exceeds 70–80%, the router often fails to find viable routing paths, leading to timing closure difficulties or even synthesis failure. A 50% utilization rate resides in a relatively safe region.

*(2) Infrastructure Logic.* Beyond MAC computation units, a neural network accelerator requires additional control logic, including finite state machines (for dataflow control and layer switching), address generators (for BRAM read/write), I/O interfaces (for communication with the quantum controller), clock management (PLLs and clock domain crossing synchronizers), and optional debug logic (e.g., ILA). These components typically consume an additional 10–20% of resources.

*(3) Academic Convention.* A 50% derating factor is a widely adopted conservative assumption in FPGA resource estimation studies. It avoids overly optimistic predictions while eliminating the need for concrete RTL implementation and synthesis verification. It should be noted that in actual hardware implementations, achievable utilization typically ranges between 40–60%, depending on the specific architectural design, clock frequency requirements, and synthesis tool optimization capabilities.

**Implications for deployability.** The 2.5× resource gap between VP1802 and VP1902 directly translates to a 2.5× difference in maximum parallelism $P_{\mathrm{max}}$ (84K vs. 211K PEs), which determines the feasibility boundary for real-time decoding. As shown in Section 4.3, this gap enables certain configurations infeasible on VP1802 to become viable on VP1902. However, even on VP1902, code distance $d=9$ represents the practical limit for single-FPGA architectures.

### D.4. HLS Synthesis Validation

To cycle-accurately validate the analytical resource estimation in Sec. 3.3, we synthesize the TCN decoder (Large, $d=9$, W4A4 QAT + $80\%$ global unstructured magnitude pruning) using Vitis HLS 2025.2, targeting the VP1902 FPGA at a $350$ MHz operating frequency. All convolutional modules achieve initiation interval II=1. The full classifier, including per-position Linear, MeanPool, LayerNorm, and final Linear, is implemented as a single top-level HLS module.

Table 14 reports the HLS synthesis result for each layer. Throughout this section, "Conv2D encoder" refers to the ResBlock together with the three subsequent plain Conv2D layers, which are pipelined as a single spatial-encoding stage. The Conv2D encoder accounts for $\sim 60\%$ of LUT consumption, the Conv1D temporal layers $\sim 24\%$, and the Readout module $\sim 16\%$. LUT cost within Readout is dominated by the fully-unrolled $128 \times 128$ per-position linear projection; the downstream MeanPool, LayerNorm, and $c_2 \rightarrow 1$ final Linear contribute negligibly.

*Table 14.* **Per-layer HLS synthesis results** for the TCN (Large) decoder on VP1902. All convolutional modules achieve II=1. Latency is reported as the range across output-channel groups within each layer.

| Layer | LUT | FF | DSP | BRAM | Latency (CC) | II |
|---|---|---|---|---|---|---|
| ResBlock ($64 \rightarrow 64$) | 286,406 | 205,355 | 128 | 0 | 106–110 | 1 |
| Conv2D-1 ($64 \rightarrow 64$) | 272,507 | 185,414 | 128 | 0 | 105–107 | 1 |
| Conv2D-2 ($64 \rightarrow 128$) | 550,879 | 383,449 | 256 | 0 | 105–109 | 1 |
| Conv2D-3 ($128 \rightarrow 128$) | 1,123,717 | 839,443 | 256 | 0 | 116–121 | 1 |
| Conv1D-1 ($128 \rightarrow 128$) | 443,451 | 227,267 | 0 | 0 | 739–741 | 1 |
| Conv1D-2 ($128 \rightarrow 128$) | 451,141 | 242,764 | 0 | 0 | 739–742 | 1 |
| Embedding | 1,126 | 297 | 0 | 64 | 724 | 1 |
| Readout | 582,310 | 312,644 | 640 | 0 | 90 | — |
| **Total** | **3,711,537** | **2,396,633** | **1,408** | **64** | — | — |

Table 15 reports the total resource utilization against the VP1902 capacity. The $43.9\%$ LUT utilization indicates that the decoder fits comfortably within the VP1902 logic fabric, leaving substantial headroom for routing and the surrounding infrastructure logic—finite state machines for dataflow control, address generators for BRAM access, I/O interfaces to the quantum controller, clock management (PLLs and clock-domain-crossing synchronizers), and debug logic, which is consistent with the $50\%$ derating factor discussed in Appendix D.3. DSP and BRAM utilization remain low ($20.5\%$ and $< 1\%$, respectively), confirming that INT4 quantization successfully shifts the arithmetic workload from scarce DSP slices to abundant LUT resources, consistent with the logic-bound regime identified in Sec. 3.3.

*Table 15.* **Resource utilization on VP1902.** The decoder fits comfortably within the VP1902 fabric; LUT is the dominant resource, while DSP and BRAM have ample headroom.

| Resource | Usage | VP1902 Total | Utilization |
|---|---|---|---|
| LUT | 3,711,537 | 8,460,000 | 43.9% |
| FF | 2,396,633 | 16,920,000 | 14.2% |
| DSP58 | 1,408 | 6,864 | 20.5% |
| BRAM | 1.15 Mb | 239 Mb | $< 1\%$ |

In continuous QEC operation, the decoder processes one new syndrome frame per measurement cycle, reusing intermediate results from previous frames via on-chip circular buffers. Table 16 reports the latency breakdown under this incremental mode, which determines real-time feasibility.

## E. Extended Numerical Results

### E.1. Understanding the Data-First Regime: From Memorization to Generalization

Sec. 4.1 established that QEC decoding operates in a data-first regime, where performance gains are driven predominantly by dataset scale rather than architectural sophistication. Here, we provide mechanistic insight into this phenomenon by

*Table 16.* **Incremental inference latency** for the TCN (Large) decoder at $d=9$ on VP1902 @ 350 MHz. The decoder processes one new syndrome frame per QEC cycle, reusing buffered intermediate results.

| Module | Clock Cycles | Latency ($\mu$s) |
|---|---|---|
| Conv2D Encoder (1 frame, pipelined) | 193 | 0.55 |
| Conv1D temporal stage (incremental) | 28 | 0.08 |
| Readout | 46 | 0.14 |
| **Total (incremental)** | **267** | **0.77** |

examining the training dynamics under varying data budgets. Specifically, we investigate how data scale fundamentally alters the learning behavior, i.e., transforming models from memorizing training examples to generalizing error patterns.

**The memorization-to-generalization transition.** Fig. 6 reveals a striking dichotomy in training dynamics at $d=7$. With limited data ($10^5$ samples), both 3D-CNN and TCN exhibit severe overfitting: training loss converges to near-zero while validation loss plateaus, yielding large train-validation gaps (0.56 and 0.17, respectively). This behavior indicates that models are *memorizing* individual syndrome-label pairs rather than extracting the underlying error correction logic.

Scaling to $5\times10^6$ samples triggers a qualitative shift. The train-validation gap collapses to $\approx 0.01$ for both architectures, demonstrating successful generalization. Critically, this transition occurs without any architectural changes or explicit regularization—**data scale alone** enables the model to cross the memorization barrier and learn robust decoding strategies. This provides a mechanistic explanation for the data-first regime: large datasets are not merely beneficial but *necessary* for neural decoders to transition from rote memorization to genuine pattern recognition.

**Overfitting vs Normal Training (d=7)**

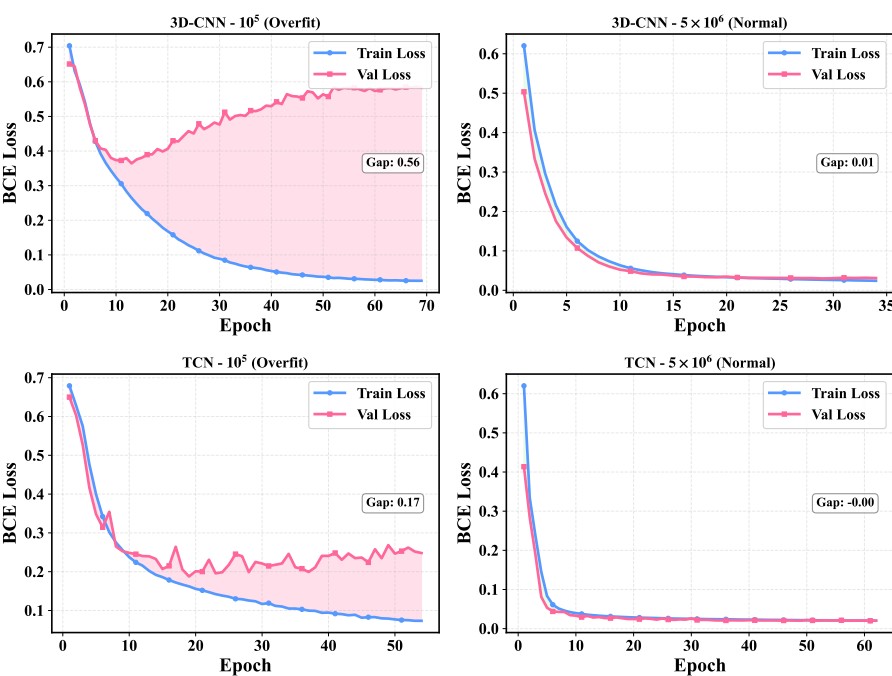

*Figure 6.* **Data scale triggers a memorization-to-generalization transition.** Training dynamics for 3D-CNN and TCN at $d=7$ under two data regimes: $10^5$ samples (left, data-scarce) vs $5\times10^6$ samples (right, data-abundant). With limited data, severe train-validation gaps emerge (0.56 for 3D-CNN, 0.17 for TCN), indicating that models memorize training examples rather than learning generalizable error patterns. Scaling to $5\times10^6$ samples eliminates this pathology: the train-validation gap collapses to $\approx 0.01$ for both architectures, demonstrating a qualitative transition to proper generalization. This phase transition occurs purely through data scaling, without any architectural modification or explicit regularization, providing mechanistic evidence for why QEC decoding operates in a data-first regime.

**Saturation of the data-first regime.** The analysis above was carried out within the fixed-dataset regime used in the main-text benchmarks (up to $5 \times 10^6$ samples for $d{=}5$ small and $2.5 \times 10^7$ samples for $d{=}7$ large; Fig. 3). A natural question is whether the data-first phenomenon continues to yield meaningful accuracy gains as the training set grows beyond this regime, or whether it eventually saturates. To answer this, we additionally train the TCN decoder with unlimited on-the-fly data generation (*infinite online training*) via the Stim sampling API (Gidney, 2021): each optimization step receives a fresh batch of independently sampled syndromes, eliminating the fixed-dataset constraint, and training continues until convergence (early stopping with patience 30). Table 17 compares offline (fixed-dataset) and infinite online training under uniform depolarizing noise at $p = 0.005$.

*Table 17.* **Offline (fixed dataset) vs. infinite online training** for the TCN decoder under uniform depolarizing noise at $p{=}0.005$. Results are reported over 3 independent runs per configuration.

| Setting | Offline LER (%) | Offline samples | Infinite LER (%) | Infinite samples | PyMatching LER (%) |
|---------|-----------------|-----------------|------------------|------------------|--------------------|
| $d{=}5$, small | $1.17 \pm 0.07$ | $5 \times 10^6$ | $\mathbf{1.03 \pm 0.04}$ | $\sim 1.0\,\mathrm{B}\ (200\times)$ | 1.42 |
| $d{=}7$, large | $0.61 \pm 0.09$ | $2.5 \times 10^7$ | $\mathbf{0.56 \pm 0.03}$ | $\sim 0.9\,\mathrm{B}\ (36\times)$ | 0.99 |

Despite consuming 36–200× more training samples, infinite online training improves the mean LER by at most $0.14\%$ (at $d{=}5$: $1.17 \to 1.03$; at $d{=}7$: $0.61 \to 0.56$), and both configurations remain well below the MWPM baseline. In contrast, the inter-run standard deviation shrinks markedly under infinite online training (most visibly at $d{=}7$, $\pm 0.09 \to \pm 0.03$), indicating that unbounded data primarily improves training *stability* rather than peak accuracy. These results confirm that the data-first regime reaches *saturation* at the dataset scales used in the main text, with further scaling yielding diminishing returns on LER and serving mainly to tighten run-to-run variance.

**Training objective reliability.** To ensure that our findings are not artifacts of metric mismatch, we verify that the training objective (BCE loss) faithfully tracks the true performance metric (LER). Fig. 7 demonstrates a consistent log-linear relationship between BCE and LER throughout training for both 3D-CNN and Transformer at $d{=}7$. Validation loss exhibits a tighter correlation than training loss, confirming it as a robust indicator of generalization. This alignment validates our training protocol: optimizing BCE effectively minimizes logical error rates, ensuring that the data-first conclusions are grounded in reliable optimization dynamics.

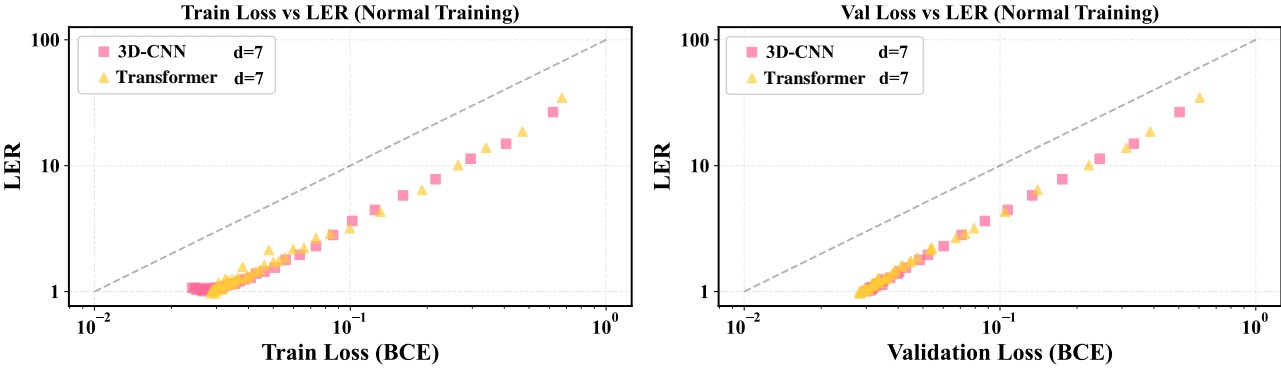

**BCE Loss vs LER Relationship (Normal Training, d=7, size=5e6)**

*Figure 7.* **BCE loss is a reliable surrogate for logical error rate.** Correlation between binary cross-entropy (BCE) loss and logical error rate (LER) during training for 3D-CNN and Transformer at $d{=}7$ with $5{\times}10^6$ samples. Both training loss (left) and validation loss (right) exhibit strong log-linear relationships with LER, confirming that minimizing the BCE objective effectively drives down the true decoding performance metric. Validation loss demonstrates tighter correlation (smaller scatter), suggesting it is a more robust predictor of generalization performance. This alignment ensures that our empirical conclusions regarding data scaling and architectural choices are grounded in a training procedure that faithfully optimizes the intended decoding objective.

### E.2. The Necessity of Inductive Bias: Architectural Scalability Analysis

While Sec. 4.1 established that QEC decoding operates in a data-first regime, this finding does not imply that architectural choice is arbitrary. Rather, the data-first observation carries an important prerequisite: the architecture must possess inductive biases that align with the inherent structure of the decoding problem. Here, we systematically investigate how different architectural priors affect scalability to larger code distances, examining both the absence of inductive bias (MLP) and the misalignment of inductive bias (GNN) as revealing counterexamples.

**Systematic comparison across architectures.**   Fig. 8 presents a comprehensive view of how four representative architectures scale with code distance $d \in \{3, 5, 7, 9\}$. The three structure-aware models—3D-CNN, TCN, and Transformer—consistently maintain logical error rates below the MWPM baseline across all distances. Notably, their performance curves track the MWPM trend, preserving a consistent margin of improvement as problem complexity increases. This robust scaling behavior stems from their embedded inductive biases: 3D-CNN's translation invariance captures the spatial regularity of the 2D lattice, TCN's temporal convolution exploits sequential error propagation, and Transformer's attention mechanism aggregates long-range correlations.

In stark contrast, the MLP exhibits a qualitatively different trajectory. At small distances ($d$=3, 5), MLP achieves comparable performance to structured models, suggesting that brute-force memorization suffices when the syndrome space is limited. However, as $d$ increases to 7 and 9, MLP's error rate escalates sharply, crossing above the MWPM baseline and reaching 3–4$\times$ higher error rates than the structured alternatives. This divergence reveals a fundamental scalability barrier: without spatial or temporal priors, the fully-connected MLP must learn the lattice geometry and error dynamics purely from data, a task that becomes intractable as the syndrome dimensionality grows cubically with $d$.

**Deep dive into MLP failure mode.**   To understand whether MLP's poor scaling can be remedied by simply increasing the training data, we conduct a detailed ablation study at $d$=7 and $d$=9. Fig. 9 tracks MLP's logical error rate as the training set scales from $10^5$ to $5\times10^6$ samples. At $d$=7, MLP demonstrates continuous improvement with data, dropping from 32% to 3.2%. However, this final error rate remains 3.4$\times$ higher than the MWPM baseline (0.93%), indicating that the model has not learned an effective decoding strategy despite access to massive training data.

The situation deteriorates further at $d$=9. Here, MLP exhibits a training plateau: error rates hover around 40% until $5\times10^6$ samples, then drop to 16.9%—still 26$\times$ worse than MWPM (0.64%). This behavior reveals a fundamental limitation: the MLP's fully-connected architecture cannot efficiently exploit the 2D spatial locality and temporal correlation inherent in surface code syndromes. As syndrome volume scales as $\mathcal{O}(d^3)$ (with $r$=$d$ rounds), the unstructured parameter space of MLP grows prohibitively large, and even large-scale data cannot guide the model to discover the underlying lattice structure.

**GNN: A case of misaligned inductive bias.**   Beyond the absence of inductive bias (MLP), architectural mismatch can also arise when the embedded prior does not align with the problem structure. GNNs exemplify this scenario. Unlike MLP, GNN possesses a clear inductive bias: iterative message passing on graph topology, designed to approximate belief propagation. This structure has proven effective for quantum LDPC codes, where Tanner graphs exhibit favorable connectivity properties.

However, on surface codes, GNN faces fundamental limitations due to the prevalence of short cycles (length-4 loops) in the stabilizer graph, which cause message oscillations during iterative decoding. Table 18 presents GNN performance at $d$=3 and $d$=5 using neural belief propagation with $5\times10^6$ training samples. Even with increased message-passing iterations (from $d$ to $2d$ rounds), GNN achieves only moderate performance: 1.76% at $d$=3 and 2.93% at $d$=5. Critically, these error rates are substantially higher than those of structure-aware models (3D-CNN, TCN, Transformer) reported in Figure 8, and even fall short of classical belief propagation combined with ordered statistics decoding (BP+OSD: 1.59% at $d$=5).

Given this poor scaling trend at small distances, we did not extend GNN experiments to $d$=7 and $d$=9, as the architecture's fundamental mismatch with surface code topology precludes competitive performance. This negative result reinforces a critical refinement: possessing *some* inductive bias is insufficient—the bias must be appropriately aligned with the problem's inherent structure.

**Implications for architectural design.**   These findings refine our understanding of the data-first regime through two distinct failure modes. MLP demonstrates that *absence* of inductive bias leads to scalability collapse, even under massive data regimes. GNN reveals that *misalignment* of inductive bias, despite being well-motivated in other contexts (e.g., qLDPC codes), similarly prevents effective scaling when the architectural prior does not match the problem structure.

**Model Performance vs Code Distance**

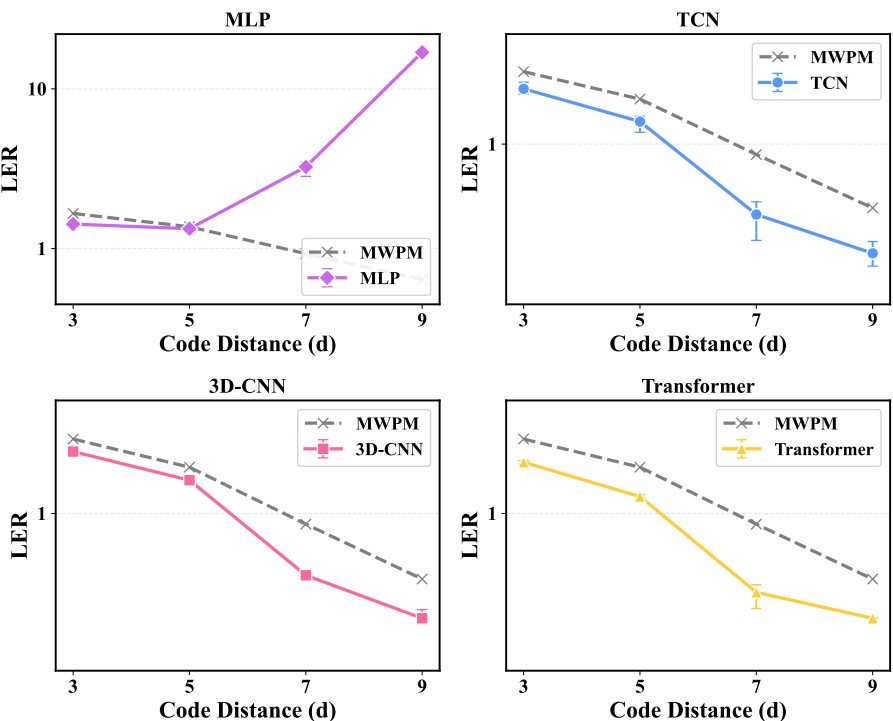

*Figure 8.* **Inductive bias determines architectural scalability to large code distances.** Performance comparison of four neural decoder architectures across code distances $d \in \{3, 5, 7, 9\}$, evaluated on $5 \times 10^6$ training samples. Structure-aware architectures (3D-CNN, TCN, Transformer) consistently outperform the MWPM baseline across all distances, with error rates tracking the baseline trend and maintaining a stable performance margin. In contrast, the structure-agnostic MLP achieves competitive performance at small distances ($d=3, 5$) but undergoes sharp degradation at $d=7, 9$, crossing above the MWPM threshold. This divergence demonstrates that embedded inductive biases—translation invariance (CNN), temporal structure (TCN), and long-range attention (Transformer)—are essential for scaling to complex decoding tasks, whereas brute-force parameterization without structural priors fails as problem dimensionality increases.

**MLP: LER vs Training Size**

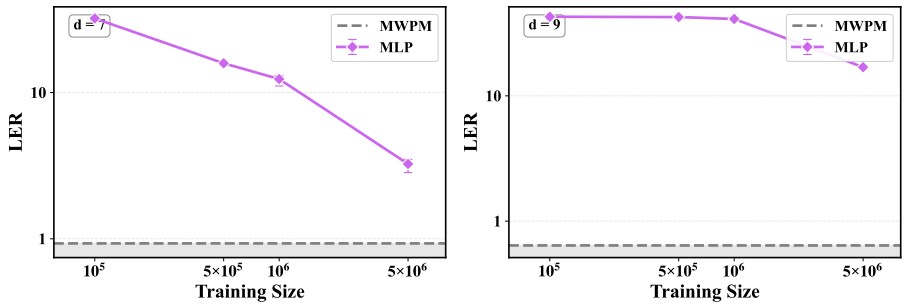

*Figure 9.* **Data scale cannot compensate for the absence of inductive bias.** MLP performance as a function of training set size at large code distances ($d=7$ and $d=9$). At $d=7$ (left), MLP improves from 32% to 3.2% LER as data scales to $5 \times 10^6$ samples, yet remains $3.4\times$ above the MWPM baseline (0.93%). At $d=9$ (right), performance plateaus near 40% until the largest dataset, then drops to 16.9%—still $26\times$ worse than MWPM (0.64%). This persistent underperformance, despite access to massive training data, demonstrates that the structure-agnostic fully-connected architecture fundamentally cannot scale to high-dimensional syndrome spaces. Without spatial or temporal inductive biases to exploit the 2D lattice geometry and sequential error dynamics, even data-intensive training fails to recover effective decoding strategies, providing a critical qualification to the data-first regime: *right inductive bias is a prerequisite, not a luxury.*

*Table 18.* **GNN-based neural belief propagation underperforms on surface codes.** Performance comparison (LER in %) at $d{=}3$ and $d{=}5$ using $5{\times}10^6$ training samples. GNN employs learnable message passing with iteration counts scaled by code distance ($d$ or $2d$ rounds). While GNN substantially outperforms vanilla belief propagation (BP), it falls short of both structure-aware neural decoders (Figure 8) and classical BP with ordered statistics decoding (BP+OSD). The poor performance at small distances, attributed to short-cycle-induced message oscillations in the surface code Tanner graph, motivated us not to pursue experiments at larger distances ($d{=}7, 9$), as the architectural mismatch precludes competitive scaling.

| Code Distance | GNN (iter=$d$) | GNN (iter=$2d$) | BP | BP+OSD |
|---|---|---|---|---|
| $d = 3$ | 1.844 | **1.758** | 9.96 | 2.05 |
| $d = 5$ | 3.450 | 2.926 | 23.64 | **1.59** |

The correct interpretation is not merely that "data scale dominates architectural choice," but rather that "*given an architecture with appropriate inductive bias aligned to the problem geometry*, data scale becomes the primary driver of performance." For surface code decoding, this means spatial locality and temporal correlation (captured by CNN and TCN) or global context aggregation (Transformer) are essential structural priors. In contrast, both graph-based message passing (GNN) and unstructured parameterization (MLP) fail to scale effectively, albeit for fundamentally different reasons: GNN's iterative message passing is disrupted by short cycles, while MLP lacks any geometric prior altogether.

This principle extends beyond QEC decoding to any spatiotemporally structured prediction task, underscoring the necessity of co-designing architectural inductive biases with problem structure rather than relying solely on data scale or model capacity.

### E.3. Training Efficiency Comparison

While previous sections focused on decoding accuracy and hardware deployability, training efficiency represents another practical consideration for neural decoder development. Figure 10 compares the training dynamics of three architectures at $d{=}9$ using $2.5{\times}10^7$ samples—the largest dataset scale required to achieve robust sub-MWPM performance.

The results reveal dramatic differences in computational cost. 3D-CNN converges to the MWPM baseline in approximately 45 minutes, making it the most training-efficient option. TCN requires roughly 80 minutes—still reasonable for iterative development. In contrast, Transformer demands approximately 500 minutes to reach comparable performance, representing more than a $10\times$ slowdown relative to 3D-CNN. Notably, all three architectures achieve similar final accuracy (LER $\approx$0.47–0.49%), indicating that the efficiency gap does not stem from convergence difficulties but rather from inherent architectural complexity.

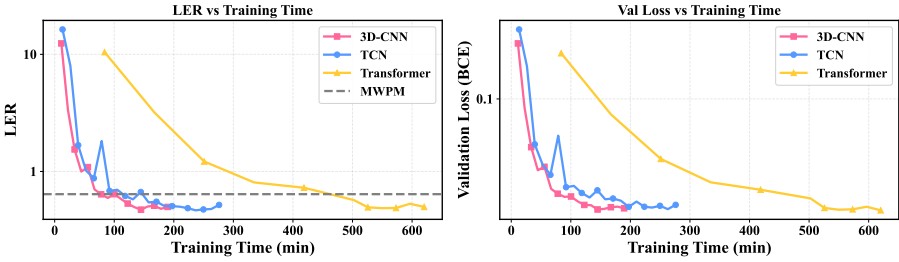

*Figure 10.* **Training efficiency varies significantly across architectures.** Training dynamics for 3D-CNN, TCN, and Transformer at $d{=}9$ with $2.5{\times}10^7$ samples. Left: Logical error rate (LER) vs wall-clock training time. Right: Validation loss vs training time. All three architectures achieve similar final performance (LER $\approx$0.47–0.49%, below MWPM baseline of 0.64%), but training time differs dramatically: 3D-CNN reaches the MWPM threshold in $\sim$45 minutes, TCN in $\sim$80 minutes, and Transformer requires $\sim$500 minutes—more than $10\times$ slower than 3D-CNN. This efficiency gap has practical implications for iterative model development and rapid prototyping in QEC decoder research.

These efficiency disparities carry direct implications for practical decoder development workflows. Fast training enables rapid prototyping, hyperparameter tuning, and architecture exploration—critical for research scenarios where multiple design iterations are necessary. In resource-constrained environments or time-sensitive deployment pipelines, training efficiency becomes a key factor in model selection, potentially outweighing marginal accuracy differences. Combined with

the hardware analysis in Sec. 4.3, these results underscore that effective neural QEC decoder design requires balancing multiple objectives: decoding accuracy, inference latency, and training cost.

## E.4. Cross Error-Rate Generalization and Noise Model Robustness

Beyond the fixed-noise-model benchmarks in Sec. 4.1, we conduct cross-error-rate generalization experiments that vary the decoder's operating conditions along two orthogonal axes: (i) the *noise model*—uniform depolarizing (as in the main text) versus the physically motivated SI1000 model (Gidney et al., 2021), which captures realistic hardware asymmetries such as dominant measurement noise—and (ii) the *test-time physical error rate*, stressed via zero-shot transfer from a single training error rate to strictly lower rates without any retraining. Crossing these two axes yields two experiments, one per noise model, each sweeping a range of physical error rates at test time. Both experiments use the TCN decoder trained to convergence via infinite online training (unlimited on-the-fly syndrome generation, as used in the saturation analysis of Appendix E.1), with the Small configuration at $d=5$ and the Large configuration at $d=7$.

**Experimental setup.** For the uniform depolarizing noise model, the training error rate is $p = 0.005$; for the SI1000 noise model, the training base error rate is $p_{\text{base}} = 0.004$, which introduces differentiated error rates across operations ($p_{\text{gate}} = p_{\text{base}}$, $p_{\text{measure}} = 5p_{\text{base}}$, $p_{\text{reset}} = 2p_{\text{base}}$, $p_{\text{idle}} = 2p_{\text{base}}$). For zero-shot evaluation, each trained model is applied directly to test sets at lower error rates without any additional training. We additionally report a lightweight fine-tuning protocol (learning rate $5 \times 10^{-5}$, 20 epochs) for reference. Test set sizes are scaled at low error rates for statistical reliability ($2 \times 10^5$ at $p=0.003$, $5 \times 10^5$ at $p=0.002$, $5 \times 10^6$ at $p=0.001$). All results are reported as mean $\pm$ std over 3 independent runs.

Tables 19 and 20 present the cross error-rate results under the uniform and SI1000 noise models, respectively.

*Table 19.* **Cross error-rate generalization under uniform depolarizing noise.** TCN trained at $p=0.005$ with infinite online data, evaluated at lower error rates via zero-shot transfer and lightweight fine-tuning. LER values (%) reported as mean $\pm$ std over 3 runs. Bold marks the lowest LER per row.

| $d$ | $p$ | TCN (Zero-shot) | TCN (Fine-tuned) | MWPM |
|---|---|---|---|---|
| | 0.005 (train) | **1.03 $\pm$ 0.04** | – | 1.42 $\pm$ 0.01 |
| | 0.004 | 0.50 $\pm$ 0.03 | **0.49 $\pm$ 0.03** | 0.76 $\pm$ 0.02 |
| 5 | 0.003 | 0.21 $\pm$ 0.01 | **0.20 $\pm$ 0.01** | 0.32 $\pm$ 0.00 |
| | 0.002 | **0.06 $\pm$ 0.00** | **0.06 $\pm$ 0.00** | 0.10 $\pm$ 0.00 |
| | 0.001 | **0.01 $\pm$ 0.00** | **0.01 $\pm$ 0.00** | **0.01 $\pm$ 0.00** |
| | 0.005 (train) | **0.56 $\pm$ 0.03** | – | 0.99 $\pm$ 0.03 |
| | 0.004 | 0.20 $\pm$ 0.02 | **0.19 $\pm$ 0.02** | 0.43 $\pm$ 0.03 |
| 7 | 0.003 | 0.06 $\pm$ 0.00 | **0.05 $\pm$ 0.00** | 0.14 $\pm$ 0.00 |
| | 0.002 | **0.01 $\pm$ 0.00** | **0.01 $\pm$ 0.00** | 0.03 $\pm$ 0.00 |
| | 0.001 | **0.00 $\pm$ 0.00** | **0.00 $\pm$ 0.00** | **0.00 $\pm$ 0.00** |

*Table 20.* **Cross error-rate generalization under SI1000 noise model.** TCN trained at $p_{\text{base}}=0.004$ with infinite online data, evaluated at lower base error rates. LER values (%) reported as mean $\pm$ std over 3 runs. Bold marks the lowest LER per row.

| $d$ | $p_{\text{base}}$ | TCN (Zero-shot) | TCN (Fine-tuned) | MWPM |
|---|---|---|---|---|
| | 0.004 (train) | **2.29 $\pm$ 0.05** | – | 2.84 $\pm$ 0.08 |
| 5 | 0.003 | 0.98 $\pm$ 0.02 | **0.97 $\pm$ 0.01** | 1.28 $\pm$ 0.02 |
| | 0.002 | 0.28 $\pm$ 0.01 | **0.27 $\pm$ 0.01** | 0.39 $\pm$ 0.01 |
| | 0.001 | **0.03 $\pm$ 0.00** | 0.04 $\pm$ 0.00 | 0.05 $\pm$ 0.00 |
| | 0.004 (train) | **1.37 $\pm$ 0.06** | – | 2.17 $\pm$ 0.05 |
| 7 | 0.003 | 0.41 $\pm$ 0.00 | **0.40 $\pm$ 0.00** | 0.72 $\pm$ 0.01 |
| | 0.002 | **0.08 $\pm$ 0.00** | **0.08 $\pm$ 0.01** | 0.14 $\pm$ 0.00 |
| | 0.001 | **0.00 $\pm$ 0.00** | 0.01 $\pm$ 0.00 | 0.01 $\pm$ 0.00 |

The results in Tables 19 and 20 show three consistent patterns. First, the TCN decoder beats MWPM in zero-shot evaluation at every tested physical error rate and under both noise models; the gap is largest at the training error rate and narrows at

lower rates as both decoders approach near-perfect accuracy. Second, the TCN–MWPM gap is larger under SI1000 than under uniform depolarizing noise ($d$=7: $0.80\%$ vs. $0.43\%$; $d$=5: $0.55\%$ vs. $0.39\%$, measured at the respective training error rates), consistent with matching-based decoders being less well suited to the asymmetric correlations introduced by dominant measurement noise ($p_{\text{measure}} = 5p_{\text{base}}$). Third, the lightweight fine-tuning pass changes the absolute LER by less than $0.02\%$ in either direction across all configurations, so a single TCN checkpoint trained at the critical regime suffices to cover the full range of tested operating points without retraining.

