# OpenReview forum: "Rethink the Role of Neural Decoders in Quantum Error Correction"
_ICML.cc/2026/Conference — ICML 2026 regular_

### Official Review · Reviewer_2xJj · 2026-02-26

**Soundness:** 2
**Presentation:** 3
**Significance:** 3
**Originality:** 2
**Overall Recommendation:** 4
**Confidence:** 3

**Summary:**

This paper revisits neural decoders for quantum error correction with a focus on accuracy–latency trade-offs under FPGA deployment constraints. The motivation is practically meaningful, and the systematic re-implementation and evaluation provide useful empirical insights, particularly regarding the importance of dataset scale and the role of quantization-aware training. However, the work primarily centers on engineering realization and benchmarking of existing architectures rather than introducing new learning methodologies. The theoretical analysis is limited. Overall, while the study is carefully executed, its methodological contribution to the machine learning community appears modest.

**Compliance With Llm Reviewing Policy:**

Affirmed.

**Final Justification:**

This work presents empirical evaluations of neural decoders for QEC and demonstrates FPGA deployment of multiple architectures, offering a important step toward hardware acceleration, and providing some good insights. Based on my evaluation of the paper, as well as the overall feedback from other reviewers, I acknowledge that the authors have conducted a comprehensive implementation and evaluation of neural decoders for QEC in hardware. That said, I still encourage the authors to further improve the organization of the paper. For example, it would be beneficial to explicitly summarize and highlight the hardware-oriented adaptations associated with each network architecture, as this was not clear during my initial reading.

**Key Questions For Authors:**

Please see Weaknesses.

**Limitations:**

yes

**Strengths And Weaknesses:**

Strengths:
The paper provides a systematic FPGA-based re-evaluation of neural decoders for quantum error correction under accuracy–latency constraints, which is a practically meaningful and well-motivated direction. By presenting empirical results, the study offers useful empirical insights, particularly regarding the importance of training dataset scale and the necessity of QAT for hardware-constrained deployment.

Weaknesses:
1. Positioning with respect to ICML: The contribution is primarily system-level evaluation. The authors should more clearly articulate what new machine learning knowledge is produced, beyond careful engineering and benchmarking.
2. The work mainly re-implements and re-evaluates existing neural decoder architectures on FPGA, with QAT/PTQ integration. While systematic, the paper does not introduce a clearly novel algorithmic or modeling contribution beyond engineering refinement.
3. Given the ICML venue, the paper would benefit from at least minimal theoretical framing: (a) The insights (e.g., dataset size > architecture; QAT is necessary) are interesting but remain largely empirical observations. The paper would be stronger if these were distilled into clearer, more general principles or guidelines supported by theoretical analysis. (b) For example, analysis or formal discussion explaining why dataset scale dominates architectural choices, or why QAT is fundamentally necessary under certain quantization regimes.

---

> ### Author Rebuttal · Authors · 2026-03-31
>
> We sincerely thank the reviewer for the thoughtful and constructive feedback. We address each concern below.
>
> ### **Response to W1 (Positioning with respect to ICML)**
> We respectfully disagree that our contribution is “*primarily engineering/benchmarking*.” In modern ML, top venues have repeatedly recognized that carefully controlled re-evaluations can be a strong ML contribution when they produce a new understanding of a practically important and previously unresolved problem. By our count, recent top-tier ML conferences included a substantial number of such papers (e.g., **28 at ICML 2025, 52 at NeurIPS 2025, and 63 at ICLR 2026**), reflecting that this contribution format is well established in the community. A canonical example is “*Understanding deep learning requires rethinking generalization*” (ICLR 2017), which advanced ML via systematic empirical insights rather than architectural novelty.
>
> More importantly, our paper **produces concrete ML knowledge, not just measurements**. We explicitly answer two long-standing questions in neural QEC decoding as indicated in the main text:
> 1. Whether reported gains stem from architectural design or from training data and supervision;
> 2. How neural decoding can be made compatible with stringent microsecond-level real-time constraints.
>
> These are not addressed by prior work that primarily reports offline accuracy.
>
> We would like to further emphasize that **the remaining three reviewers recognized these contributions as substantive**: Reviewer 1 highlights our controlled cross-architecture evaluation that jointly considers accuracy, training cost, and hardware latency; Reviewer 2 emphasizes our deployment-feasibility evaluation that incorporates realistic hardware bottlenecks beyond logical cycles; and Reviewer 3 underscores our evidence on the necessity of inductive bias for scaling with data.
>
> We hope these clarifications help the reviewer better assess the contributions and potential impact of our submission.
>
> ### **Response to W2 (Incremental evaluation contribution)**
> We respectfully disagree that our contribution is limited to “*engineering refinement*.” As mentioned in prior response, our goal is not to propose yet another architecture, but to resolve a practically important and previously unresolved ML question in neural QEC decoding: *how decoding accuracy interacts with data scale, inductive bias, and microsecond-level deployability constraints*. To this end, we make **three methodological contributions** beyond re-implementation.
>
> First, we provide a controlled and cross-architecture study that isolates inductive bias effects under matched capacity, and identify a **QEC-specific data-first regime**. That is, once inductive bias is appropriate, decoding performance largely converges across architectures in the high-data regime. This is not a generic scaling restatement; it directly changes the design priority for AI-for-QEC from “ever more complex models” to data infrastructure and deployability-aware design (see Sec. 4.1).
>
> Second, we contribute **a hardware-aware evaluation methodology that couples decoding accuracy with real-time constraints**, including an end-to-end INT4 compression pipeline and ablations showing that QAT is necessary to preserve fidelity at INT4 (Fig. 4(a)), whereas PTQ (especially for recurrent designs) can fail catastrophically. This yields generalizable insights on quantization-resilient vs. quantization-fragile inductive biases for spatiotemporal decoding.
>
> Third, we develop a **vendor-agnostic resource/latency projection framework** that turns decoder MAC/parameter footprints into FPGA feasibility limits, revealing a concrete deployability boundary (e.g., d=9 as the single-card feasible limit under microsecond constraints) and clarifying why certain architectures cross the hardware-scaling boundary despite competitive offline accuracy (see Sec. 3.3).
>
> All of these methodological contributions are generalizable beyond our experimental setup and establish a principled framework for evaluating neural decoders under real-time constraints.
>
> ### **Response to W3 (Theoretical analysis)**
> Thanks for the comments. As noted in our response to **W1**, ICML and related venues frequently recognize impactful “rethinking/revisiting” works whose contributions are systematic empirical insights and actionable principles, even without new theoretical guarantees. A key reason is that existing theory often does not provide tight, practice-predictive characterizations for modern deep models, **making careful empirical studies a complementary route to understanding**. In this spirit, our work resolves a practically important and previously underexplored question in neural QEC decoding under microsecond-scale constraints.
>
> To address the reviewer’s concern, we will add a brief discussion in the updated version connecting our observations to standard learning-theoretic intuitions (e.g., bias–variance), without claiming new theoretical guarantees.

---

> > ### Author Rebuttal · Reviewer_2xJj · 2026-04-01
> >
> > Thank you for the authors’ clarification. I appreciate the responses regarding W1 and W3, and I find the motivation and experimental setup to be clearly presented. However, the concerns related to W2, which arose during my reading of the paper, remain unresolved even after considering the authors’ response. The authors emphasize that their work goes beyond a simple re-implementation and claim to “make three methodological contributions beyond re-implementation.”
> >
> > That said, some of these contributions appear to rely on broadly established practices. For example, the QEC-specific data-first regime (primarily in terms of scaling) aligns with general consensus in learning theory. Similarly, the hardware-aware quantization capability seems to build upon existing tools and standard approaches (e.g., the Brevitas library referenced in Appendix C.3, and conventional QAT methods).
> >
> > To summary, I believe the authors need to substantially improve the presentation of their work, in particular by more clearly organizing and emphasizing their novel contributions. At its current stage, the paper gives the impression of combining existing architectures and methods rather than introducing sufficiently distinct innovations. I therefore keep my current score.

---

> > > ### Author Response · Authors · 2026-04-02
> > >
> > > We thank the reviewer for the thoughtful follow-up and appreciate that **W1** and **W3** are now clear.
> > >
> > > Regarding **W2**, we respectfully disagree that *the contribution should be judged primarily by whether we introduce a new architecture or a novel quantization algorithm*. ICML and other top ML venues regularly accept rethinking/revisiting papers when they resolve an unresolved practical question with controlled evidence, rather than architectural novelty. A canonical example is “*Understanding deep learning requires rethinking generalization*” (ICLR 2017), which **introduced no new training algorithm or architecture**, yet fundamentally changed how the community interprets generalization.
> > >
> > > In this spirit, we reiterate that our core contribution is the rethinking result: **a controlled and hardware-relevant study of neural QEC decoding under microsecond-scale constraints**. Concretely, our experiments uncover a counterintuitive yet practically important finding:
> > > > **AlphaQubit (Nature 635, 834–840 (2024)), an archetypal “new method” approach that motivates Transformer necessity and heavy architectural design, does not retain a clear advantage under real-time constraints.**
> > >
> > > This illustrates that **impactful ML contributions need not hinge on proposing new architectures; systematic empirical re-evaluation can be equally influential when it overturns incorrect assumptions and refocuses design priorities**.
> > >
> > > In addition, our statement about “three methodological contributions beyond re-implementation” was intended as a side contribution of independent interest, not the main novelty claim; we will rephrase this to avoid giving the impression that our contribution hinges on tool-level engineering.
> > >
> > > With this perspective in mind, we address the reviewer’s remaining concerns point by point below.
> > >
> > > $\underline{\text{Comments: QEC-specific data-first regime ... with general consensus in learning theory.}}$ Our “*QEC-specific data-first regime*” is not a restatement of generic scaling folklore. In the QEC decoding community, the prevailing prior has often been close to the **opposite**: a “model-first” view that emphasizes increasingly sophisticated architectures.
> > >
> > > A concrete example is AlphaQubit, which argues that global attention is necessary to capture long-range correlations between distant detection events, and motivates Transformers over CNNs due to CNNs’ limited local receptive fields. It further introduces substantial architectural refinements (e.g., positional encoding design and recurrent mechanisms) to support this premise.
> > >
> > > **Our experimental results directly challenge this assumption**. As shown in Fig. 3 (a),  even simple and “brute-force” convolutional models, when trained with sufficient data ($>10^7$), can match or exceed these carefully engineered Transformer-based decoders. This suggests that surface-code decoding exhibits structural properties distinct from standard long-range dependency tasks, and that long-range correlations do not necessarily require a Transformer-only solution. Clarifying and correcting this design priority is precisely **the kind of field-level insight that “rethinking” papers are meant to provide**.
> > >
> > > $\underline{\text{Comments: The hardware-aware ... and standard approaches.}}$ As with other "rethinking and revisiting" papers, the key contribution here is not the tool itself, but the QEC-specific finding enabled by a controlled evaluation. That is, under the extremely accuracy-sensitive QEC setting, aggressive INT4 quantization is not “plug-and-play.”
> > >
> > > More specifically, as exhibited in Figure 4 and Table 2, we show (i) post-training quantization (PTQ) can catastrophically break certain decoder architectures (notably recurrent/gated designs), and (ii) quantization-aware training (QAT) is required to retain decoding fidelity at INT4. To our knowledge, **prior neural-decoder work has not provided a clear and reproducible report of this INT4 failure mode and the corresponding QAT necessity under real-time constraints**. Using a simple and standard QAT pipeline strengthens the claim: even the most basic approach suffices to preserve QEC-level fidelity, and more advanced methods would only further improve efficiency.
> > >
> > > $\underline{\text{Comments: I believe ... contributions}}$. Based on the clarifications above, we hope it is now clear that our work is intended as a rethinking paper: its primary contribution is to deliver systematic insights that correct prevailing assumptions and clarify design priorities for neural QEC decoding, rather than to introduce a new architecture or algorithm. **We believe our work is structured accordingly, with the main text organized around the two unresolved questions (**Q1/Q2** in Page 1) and the resulting takeaways**.
> > >
> > > We respectfully ask the reviewer to re-evaluate the quality of our work under this contribution style, and we are happy to provide any further clarification if helpful.

---

### Official Review · Reviewer_Yd1n · 2026-03-09

**Soundness:** 3
**Presentation:** 2
**Significance:** 3
**Originality:** 2
**Overall Recommendation:** 4
**Confidence:** 3

**Summary:**

This paper investigates the feasibility of real-time surface-code neural decoding by evaluating five representative architectures under accuracy-latency constraint. The authors reveal that performance is primarily driven by data scale and inductive bias rather than architectural complexity, and establishes that Quantization-Aware Training (QAT) is essential for meeting the rigorous timing demands of FPGA deployment on superconducting processors.

**Compliance With Llm Reviewing Policy:**

Affirmed.

**Final Justification:**

I'm happy to see the physical FPGA validation in the author's rebuttal, which improves the soundness of this paper. Overall, I think the empirical study in this paper is helpful enough for QEC practitioners that I'm leaning towards acceptance, but I don't think the paper has made enough contribution to warrant a higher score, so I'll maintain my score of "4: Weak accept".

**Key Questions For Authors:**

1. Are there any other publicly-available datasets other than Sycamore datasets that you can use to verify your claims?
1. Suppose someone wants to use another quantum processor hardware other than Google Sycamore, do they need to collect millions of samples from scratch, or can they somehow reuse the Sycamore dataset? How practical/costly would that be?
1. I don't think BatchNorm Folding is technically PTQ, because you can fuse layers to reduce parameter count without quantizing anything. Would this affect the conclusion of the PTQ vs QAT ablation?
1. Do you think we can meet the strict timing demands by using more advanced PTQ algorithms such as [GPTQ](https://arxiv.org/abs/2210.17323) or [QTIP](https://arxiv.org/abs/2406.11235)? The same question goes for advanced QAT algorithms such as [AWQ](https://arxiv.org/abs/2306.00978) and [SmoothQuant](https://arxiv.org/abs/2211.10438).

**Limitations:**

No. I think the authors should at least acknowledge the limitation that they are relying on an analytical resource estimation model with a 50% logic derating factor, but didn't implement it on physical FPGA, so there could be some unforeseeable issues that prevent it from used for quantum error correction on actual hardware.

**Strengths And Weaknesses:**

**Strengths**:
1. [Soundness] Figure 3 supports the claim that QEC decoding is fundamentally data-driven, given an architecture with appropriate inductive bias.
1. [Soundness] Figure 4 supports the necessity of INT4 QAT (as opposed to PTQ) for meeting real-time latency constraints.
1. [Soundness] The five architectures are representative, but also adapted to QEC (e.g., exclusion of pooling layers in CNN, etc.)
1. [Significance] Reveals three distinct feasibility regimes for the compressed neural decoders, providing practical guidelines.
1. [Significance] Emphasizes the importance of architectural inductive bias, without which the model won't scale with data.
1. [Originality] Focus on a systematic analysis under explicit accuracy–latency constraints.

**Weaknesses**:
1. [Soundness] Used a resource estimation with 50% derating factor, but didn't use actual FPGA hardware.
1. [Soundness] Only evaluated on one dataset (Sycamore).
1. [Soundness] The authors' model choice, while sensible to me, isn't provably optimal, e.g., maybe there exists a better transformer-based model that can provide better accuracy–latency tradeoff.
1. [Significance] The PTQ and QAT algorithms used here looks naive.
1. [Presentation] It's a little contradictory that the authors conclude "decoding performance is largely architecture-agnostic
in the high-data regime", but also emphasis the importance of having the right inductive bias. The sentence `The correct interpretation is not merely that “data scale dominates architectural choice,” but rather that “given an architecture with appropriate inductive bias aligned to the problem geometry, data scale becomes the primary driver of performance.”` should probably be moved from the appendix into the main text as a clarification.

---

> ### Author Rebuttal · Authors · 2026-03-31
>
> We sincerely thank the reviewer for the constructive feedback. Below, we address each concern.
>
> # W1&Limitation (Absence of physical FPGA validation)
> We have performed HLS synthesis via Vitis HLS for the TCN decoder at d=9 (large, 80% prune), targeting the VP1902 FPGA.
>
> As shown below, HLS synthesis reports a latency of 271 clock cycles. With a 350 MHz target frequency, this corresponds to ~0.77 μs per round, meeting the 1 μs deadline. Notably,  this is 19% fewer cycles than our Table 2 estimate, confirming our analytical result is conservative.
>
> |  | Analytical Estimate (Table 2) | HLS Synthesis (Vitis) |
> | --- | --- | --- |
> | Platform | VP1902 | VP1902 |
> | Model | TCN d=9, large, 80% prune | TCN d=9, large, 80% prune |
> | Incremental latency | 334 CC | ~271 CC |
> | Latency @350MHz | 0.95 μs | **0.77 μs** |
>
> We will add Vivado place-and-route results to confirm timing closure and frequency; meanwhile, HLS synthesis already provides cycle-level validation beyond analytical estimates.
>
> # W2&Q1 (Limited dataset diversity)
> Public large-scale real-device QEC datasets are extremely limited; at this scale, the Sycamore DEM is the only openly available superconducting dataset (Nature 614, 676–681).
>
> We would also highlight that to pursue diversity, our work has evaluated neural decoders **extensively under multiple synthetic noise models** (Sec 3.3).
>
> To further address this issue, we have appended new experiments under the SI1000 noise model and zero-shot cross-error-rate inference.
>
> **SI1000 noise model ($p_{base}=0.004$), TCN decoder:**
>
> | Setting | LER_TCN (%) | LER_PyMatching (%) | Relative Improvement |
> | --- | --- | --- | --- |
> | d=5, small | 2.29 ± 0.05 | 2.84 ± 0.08 | 19.4% |
> | d=7, large | 1.37 ± 0.06 | 2.17 ± 0.05 | 36.9% |
>
> **Zero-shot cross-error-rate inference (Uniform, trained at $p=0.005$):**
>
> | Target $p$ | d=5 LER_TCN (%) | d=5 LER_PyMatching (%) | d=7 LER_TCN (%) | d=7 LER_PyMatching (%) |
> | --- | --- | --- | --- | --- |
> | 0.005 | 1.03 ± 0.04 | 1.42 ± 0.01 | 0.56 ± 0.03 | 0.99 ± 0.03 |
> | 0.004 | 0.50 ± 0.03 | 0.76 ± 0.02 | 0.20 ± 0.02 | 0.43 ± 0.03 |
> | 0.003 | 0.21 ± 0.01 | 0.32 ± 0.00 | 0.06 ± 0.00 | 0.14 ± 0.00 |
> | 0.002 | 0.06 ± 0.00 | 0.10 ± 0.00 | 0.01 ± 0.00 | 0.03 ± 0.00 |
> | 0.001 | 0.01 ± 0.00 | 0.01 ± 0.00 | 0.00 ± 0.00 | 0.00 ± 0.00 |
>
> **Zero-shot cross-error-rate inference (SI1000, trained at $p_{base}=0.004$):**
>
> | Target $p_{base}$ | d=5 LER_TCN (%) | d=5 LER_PyMatching (%) | d=7 LER_TCN (%) | d=7 LER_PyMatching (%) |
> | --- | --- | --- | --- | --- |
> | 0.004 | 2.29 ± 0.05 | 2.84 ± 0.08 | 1.37 ± 0.06 | 2.17 ± 0.05 |
> | 0.003 | 0.98 ± 0.02 | 1.28 ± 0.02 | 0.41 ± 0.00 | 0.72 ± 0.01 |
> | 0.002 | 0.28 ± 0.01 | 0.39 ± 0.01 | 0.08 ± 0.00 | 0.14 ± 0.00 |
> | 0.001 | 0.03 ± 0.00 | 0.05 ± 0.00 | 0.00 ± 0.00 | 0.01 ± 0.00 |
>
> The above results show that neural decoders (i) generalize across different noise models; (ii) training at the critical regime yields zero-shot generalization to lower error rates, confirming that our conclusions are not artifacts of a single noise model or dataset.
>
> # Q2 (Cross-hardware generalization and retraining cost)
> The procedure for deploying a neural decoder on a new device (without large-scale data collection) is as follows:
> 1. **Calibration**: Collect device shots and fit a DEM.
> 2. **Data generation**: Use Stim with the DEM to generate unlimited training data.
> 3. **Training**: Pretrain on simulated data, then fine-tune on limited real shots. Table 1 shows the DEM as the dominant factor.
>
> # W3&W4&Q4 (Model optimality and quantization concerns)
> **Model optimality**. Our goal is not to identify a single “best” model. Instead, we systematically explore how different inductive biases behave under realistic QEC constraints. Our key finding is that under sufficient data, architectural differences largely converge to comparable performance, indicating a QEC-specific data-first regime.
>
> **Quantization methodology**. While GPTQ/AWQ target GPU-style mixed-precision inference, our deployment requires sub-microsecond latency on FPGAs, where uniform INT4 is a deployability requirement. We observe that performance is more sensitive to data/inductive bias than to quantization sophistication, and will note FPGA-native extensions of sensitivity-aware quantization as future work.
>
> # W5 (Data-first regime)
> We agree with the reviewer's suggestion and will revise that architectural choices matter for providing the right inductive bias, while data scale dominates once that condition is satisfied.
>
> # Q3 (BatchNorm (BN) folding and PTQ)
> Thanks for the comment. We clarify that BN folding is an inference-graph optimization applied before quantization (fusing BN into preceding layers) and is not a quantization method. To this end, we apply it identically across FP32, PTQ, and QAT to keep the inference graph consistent.
>
> Fig. 4(a) isolates the quantization strategy itself, and the conclusion that QAT is necessary to preserve INT4 accuracy is unaffected by how BN folding is categorized.

---

> > ### Author Rebuttal · Reviewer_Yd1n · 2026-04-03
> >
> > Thanks for the detailed and helpful rebuttal, especially with the physical FPGA validation. Overall, I think the empirical study in this paper is good enough that I'm leaning towards acceptance, but I don't think the paper has made enough contribution to warrant a higher score, so I'll maintain my score of "4: Weak accept".

---

> > > ### Author Response · Authors · 2026-04-05
> > >
> > > We thank Reviewer Yd1n for the thoughtful reassessment, and we appreciate that the physical FPGA validation addressed the soundness concerns.
> > >
> > > We also appreciate the reviewer’s recognition of the value of this work in its intended “rethinking/revisiting” contribution style.
> > >
> > > Thank you again for your time and consideration.
> > >
> > > Best regards,
> > >
> > > The authors

---

### Official Review · Reviewer_uNpU · 2026-03-13

**Soundness:** 3
**Presentation:** 3
**Significance:** 2
**Originality:** 2
**Overall Recommendation:** 4
**Confidence:** 4

**Summary:**

This paper presents a systematic empirical study of neural network decoders for surface-code QEC, addressing two key questions:
- Does the performance improvement of neural network decoders stem from architectural design or from the size of the training data?
- How can neural network decoders be made compatible with microsecond-order real-time delay constraints?

The authors integrate five representative neural network decoder architectures MLP, 3D-CNN, TCN, Transformer, and GNN into a common framework, conduct extensive benchmarking experiments on rotational surface codes up to code distance d=9, and develop an end-to-end compression pipeline QAT + pruning targeting FPGA hardware deployment. The key findings are: (i) near-term decoding performance is primarily determined by dataset size rather than architectural complexity, (ii) appropriate inductive bias tailored to the surface code topology is still essential for scalability, and (iii) INT4 quantization via QAT is a prerequisite for meeting real-time delay constraints on FPGAs.

**Compliance With Llm Reviewing Policy:**

Affirmed.

**Final Justification:**

The authors’ final clarification convincingly reframes the data-first observation as a significant rethinking of the prevailing architectural doctrine in neural QEC decoding. When combined with the high-impact validation of INT4 quantization and real-time FPGA latency constraints, the work provides a technically sound and actionable framework that justifies a move to a weak accept.

**Key Questions For Authors:**

1. How can neural network decoders be made compatible with microsecond-level real-time delay constraints? What are the author's specific contributions? Are there any new methodological contributions beyond applying existing techniques such as QAT and weight pruning to QEC?
2. What is the performance difference between the redesigned architecture in this paper and the original reference model? Can the effectiveness of the redesign be quantitatively demonstrated?
3. The conclusion "data size matters more than architecture" was drawn within a limited training data size range of 5×10⁵ to 2.5×10⁷. Does the same conclusion hold for larger data sizes beyond this range? Or, at what point does convergence occur as data size increases beyond this range?
4. Is the data-first regime independent of the noise model? The main experiments rely solely on the standard depolarizing model (p=0.005). It remains unclear whether the same conclusion holds under different noise models such as SI1000, or at different error rates beyond p=0.005.

**Limitations:**

- In Table 2, the parameters $d$, scale, and prune represent the experimental environment of the model. Therefore, in order to conduct a clear and fair comparison of model performance, it is essential to align these three conditions across models. For instance, the configuration used for 3D-CNN (d = 3, scale = small, prune = 90%) does not appear with the same environment for TCN or Transformer.
- When discussing model complexity (scale), the paper categorizes models into two groups: small and large. Models with relatively higher complexity are labeled as large, while those with lower complexity are labeled as small. However, the paper does not demonstrate consistent criteria for this relative classification. While it is natural that 3D-CNN, TCN, and Transformer have different architectural designs and different numbers of parameters, dividing them into small and large based on subjective criteria is not appropriate. Even when attempting to infer consistency through Table 2, discrepancies appear. For example, the memory size difference between 3D-CNN (3, small, 90%) and 3D-CNN (7, large, 90%) is approximately 60×, whereas the difference between TCN (3, small, 80%) and TCN (7, large, 80%) is only about 4×.
- For d < 9, 3D-CNN shows significantly higher performances than TCN when comparing the same d in terms of estimated CC. However, at d = 9, TCN becomes the only architecture capable of real-time operation. The paper briefly explains this phenomenon with the statement: “Notably, only the TCN demonstrates continued viability on the high-end VP1902 (Tcc = 334 CCs).” However, this explanation is insufficient. A more detailed discussion is necessary to clarify why the performance of the 3D-CNN model drops relative to TCN at d = 9.
- The paper states that the five architectures were selected to span the dominant inductive biases in modern supervised decoding, but does not provide explicit justification for why these five specifically constitute a representative set.

**Strengths And Weaknesses:**

* Soundness

•	The comparison between QAT and PTQ provides empirical evidence supporting the necessity of QAT for real-time deployment environments.

•	The system incorporates latency caused by hardware bottlenecks in addition to logical clock cycles, enabling a more realistic evaluation of deployment feasibility.

•	Validation using real-world Google Sycamore data strengthens the credibility of the simulation-based results.

•	The experiments with GNNs are terminated at d = 3 and d = 5, citing poor performance trends. However, without testing modified GNN variants, for example, using spatiotemporal detector graphs, it is not fully justified to conclude that the limitations are structural.

•	The entire study evaluates performance only at single noise rate p = 0.005 near the pseudo-threshold. There is no analysis of behavior below or above the threshold.

•	The Union-Find decoder, which has near-linear complexity and is often suitable for real-time decoding, is not included as a baseline comparison.

* Presentation

•	The paper provides quantitative hardware-level evaluation, including actual clock cycle estimation (Table 2), which helps clearly present practical deployment feasibility.

•	In Figure 7, a metric labeled RECON appears but is not explained in the main text.

•	The definition and criteria of the data-first regime are not clearly presented.

* Significance

•	The work addresses an important challenge in real-time QEC decoding, particularly by considering hardware constraints such as FPGA latency and clock cycles.

•	The analysis of inductive bias provides QEC-specific insight, particularly in explaining the failure modes of MLP and GNN in terms of the structural properties of surface codes.

•	The data-first regime and the memorization-to-generalization transition are essentially well-known phenomena in the deep learning literature, and the paper does not sufficiently discuss how these observations are distinct from general deep learning scaling phenomena.

* Originality

•	Individual components such as neural network-based QEC decoding, model quantization, and FPGA deployment have all been explored in prior works. The paper does not sufficiently explain what new methodological contribution arises from combining these components.

---

> ### Author Rebuttal · Authors · 2026-03-31
>
> # Response to Significance&Originality&Q1&Q2 (Novelty and contribution)
> We respectfully disagree that ML originality requires new architectures. For example, “Understanding deep learning requires rethinking generalization” (ICLR 2017) advanced ML via systematic empirical insights. In the same spirit, we systematically study neural QEC decoders under explicit microsecond-scale latency constraints. Our results show that (1) With enough data, architectures converge; and INT4 decoders beat MWPM within ($<1 \mu s$) on FPGAs.
>
> Regarding Q2, our redesigns are constraint-driven and quantitatively validated (Sec. 4.2): we remove CNN pooling to preserve spatial resolution for error localization, and replace LSTMs/GRUs with pure 1D convolutions in TCNs to avoid sigmoid/tanh saturation under low-bit quantization. This yields a strictly better accuracy–latency tradeoff than the original reference implementations: recurrent PTQ variants collapse at INT4 (Fig. 4a), whereas our redesigned TCN remains quantization-resilient, retaining near-FP32 accuracy under INT4 QAT while remaining FPGA-viable.
>
> # Response to Q3 (Data-first conclusion)
> We have appended new experiments with **infinite online data generation** (i.e., unlimited i.i.d. samples generated on-the-fly), training the TCN decoder to full convergence. Each configuration is repeated 3 times:
>
> | Setting | Offline LER (%) | Offline samples | Infinite LER (%) | Infinite samples | PyMatching LER (%) |
> | --- | --- | --- | --- | --- | --- |
> | d=5, small | 1.17 ± 0.07 | $5 \times 10^6$ | 1.03 ± 0.04 | ~1.0B (200×) | 1.42 |
> | d=7, large | 0.61 ± 0.09 | $2.5 \times 10^7$ | 0.56 ± 0.03 | ~0.9B (36×) | 0.99 |
>
> The implications are:
> 1. Saturation. Despite a 36–200× increase in training data, mean LER improves by at most 0.14%. This implies that further data scaling yields diminishing returns, consistent with Fig. 3(a).
> 2. Stability. The standard deviation under infinite data is much smaller (d=7: ±0.03 vs ±0.09), showing that infinite online data allows more robust convergence.
>
> # Response to Sound5&Q4 (Noise-model generalization)
> The choice of $p = 0.005$ follows the convention and is the most challenging one, not arbitrary convenience, as noted in our response to Reviewer gg4N (W1).
>
> Additional experiments (Reviewer Yd1n, **W2&Q1**) validate our claims: training at p = 0.005 enables perfect zero-shot generalization to lower error rates; the data-first regime holds across different noise models.
>
> # Response to Sound6 (UF decoder)
> The UF decoder, while efficient, is typically less accurate than MWPM. Since our goal is to satisfy microsecond-scale latency while exceeding MWPM-level accuracy, we use an SOTA MWPM as our primary baseline. In doing so, any neural decoder with a lower LER than MWPM under the same latency constraints would also surpass UF in accuracy.
>
> # Response to Sound4 (Evaluation of GNN variants)
> A spatiotemporal-grid GNN adds no new inductive bias: local message passing reduces to 3D-CNN/TCN, while global connectivity resembles Transformer attention; since we already evaluate both, this variant is largely redundant.
>
> More broadly, our goal is not to crown a “best” architecture, but to show that with sufficient data, performance largely converges across architectures.
>
> # Response to L1/L2/L3 (fairness and scaling)
> We ensure fair comparison by aligning all models with the same hidden dimension (feature size per detector), the principled capacity metric for surface-code decoding; parameter-count differences therefore reflect architectural scaling rather than inconsistent settings. Pruning ratios are also non-arbitrary: each is set to the maximum sparsity that preserves sub-MWPM performance (Fig. 4(b)), since a uniform ratio would under-prune robust architectures or over-prune fragile ones and distort deployability.
>
> “Small/large” corresponds to two shared hidden-dimension settings across architectures. The observed parameter-growth disparity (e.g., 3D-CNN vs. TCN) follows distinct scaling laws: 3D-CNN’s spatiotemporal flattening yields $O(d^3)$ growth, whereas TCN’s separated temporal–spatial processing yields $O(d^2)$.
>
> Accordingly, the d=9 crossover reflects a hardware-scaling boundary, not an accuracy drop: 3D-CNN is more accurate for d<9 within FPGA limits, but its cubic MAC/LUT growth exceeds microsecond-level capacity at d=9, while TCN remains viable due to quadratic growth.
>
> # Response to L4
> They are representative because they span the dominant spatiotemporal inductive biases in neural QEC decoding (MLP/CNN/TCN/Transformer/GNN), under which most prior neural decoders fall as instances or minor variants.
>
> # Response to Presentation
> Fig. 7 metric: “ReCon” was an internal code name for the Transformer and will be corrected in the revision.
>
> Data-first regime: We will clarify in the main text that it means, given an architecture with appropriate geometry-aligned inductive bias, data scale becomes the primary driver of performance (moved from the appendix).

---

> > ### Author Rebuttal · Reviewer_uNpU · 2026-04-03
> >
> > Thank you for the detailed responses. The additional experiments are helpful in addressing some of my concerns regarding the paper’s empirical soundness. However, even after rethinking the work in light of these responses, I still have concerns regarding its overall contribution.
> >
> > ## Additional experiments
> >
> > The additional experiments utilizing the SI1000 noise model and the cycle-level validation through HLS synthesis successfully address the technical concerns regarding the paper’s empirical soundness.
> >
> > ## Absence of fundamental re-evaluation
> >
> > Despite these improvements, the contribution of this paper appears to be a collection of useful conclusions derived from systematic observations rather than the discovery of fundamental mechanisms or foundational principles.
> > While these observations provide practical guidance for the QEC community, they lack the profound conceptual shift or the unveiling of fundamental truths.
> > "Rethinking" papers, such as the ICLR 2017 study on generalization, gained their influence by using empirical results to expose fundamental flaws in existing theoretical frameworks.
> > In contrast, the "data-first regime" presented here is an empirical scaling observation that largely aligns with general deep learning consensus, rather than a re-evaluation of models based on fundamental QEC or learning-theoretic principles.

---

> > > ### Author Response · Authors · 2026-04-04
> > >
> > > We appreciate the reviewer for acknowledging that the SI1000 experiments and HLS synthesis address the empirical soundness concerns.
> > >
> > > Regarding the concern that our work lacks a ‘fundamental’ re-evaluation, we note that this point was newly raised in the second round. Below, we address it by clarifying our rethinking contribution and its fundamental significance for QEC decoding.
> > >
> > > 1. In the ‘Rethinking’ Spirit of ICLR 2017 and Related Revisiting Works.
> > >
> > > >As we noted in our earlier response to Reviewer 2xjj, we cited the ICLR 2017 ‘rethinking generalization’ work as an example of how top venues value impactful rethinking studies driven by systematic empirical insight. That is, such papers are often “fundamental” not because they introduce a new algorithm, but because **they identify and correct a prevailing, yet misguided—research trajectory**. A recent example in our domain is “Rethink the Role of Deep Learning towards Large-scale Quantum Systems” (ICML 2025), which shows that prior DL-vs-ML conclusions were often confounded by unfair quantum-resource assumptions in dataset construction, and re-evaluates the necessity of DL under matched resources.
> > >
> > > >Our work is likewise fundamental in the context of neural QEC decoding. Prior to our study, progress in the field has largely been guided by a community-level architectural doctrine. Concretely, a prevailing “model-first” premise, made explicit in AlphaQubit, is that long-range correlations in QEC require global attention and heavy Transformer-centric design because CNN receptive fields are allegedly insufficient. This premise has driven substantial architectural engineering, e.g., AlphaQubit variants that combine Transformers with dilated CNNs and recurrent components, and further escalate with RoPE, RMSNorm, SwiGLU, and cross-attention readouts. Here, our controlled high-data evaluation re-examines this trajectory and shows that such complexity is often unnecessary: with proper data scaling, simple convolutional decoders can match or exceed these heavily engineered models under deployment constraints. **Correcting this design priority constitutes a meaningful conceptual shift for neural QEC decoder design**.
> > >
> > > 2. Impact on the QEC Physics Community
> > >
> > > >We acknowledge that at a high level, “more data helps” aligns with broad ML intuition. However, **in neural QEC decoding, this behavior was neither established nor widely expected**. As mentioned above, the prevailing prior has been “model-first,” explicitly arguing that long-range correlations require global attention and heavy Transformer-centric design (e.g., AlphaQubit). Whether data scaling can largely close architectural gaps under realistic constraints was therefore an open QEC-specific question, not a settled consensus.
> > >
> > > >Our contribution is to **provide, for the first time, strong controlled evidence resolving this question via a large-scale cross-architecture study (≈5000 GPU-hours) with matched capacity and deployment-aware evaluation**, showing that once inductive bias is appropriate, performance converges in the high-data regime.
> > >
> > > >In words, because neural QEC decoding is an inherently interdisciplinary field, making ML scaling insights concrete and verifying that they hold for QEC under real-time constraints helps bridge the physics–ML gap and advance the field.
> > >
> > > 3. Contributions Beyond the ‘Data-First’ Regime.
> > >
> > > >Beyond our systematic re-evaluation of the field’s “model-first” trajectory in QEC, our quantization results uncover **deployment-critical mechanisms that are invisible in offline benchmarks**. In the QEC regime where $10^{-3}-10^{-4}$ differences in LER matter, INT4 quantization is not “plug-and-play”: **PTQ can fail catastrophically for recurrent/gated motifs, whereas QAT is necessary to preserve decoding fidelity**.
> > >
> > > >Taken together, these findings constitute **a field-level re-evaluation**. That is, they challenge the widely adopted assumption of Transformer necessity and establish actionable, deployment-relevant principles for real-time neural QEC decoding.
> > >
> > > Based on the above clarifications, we hope it is now clear that this submission is best evaluated in the “rethinking/revisiting” paradigm: it provides controlled and hardware-relevant evidence that corrects a prevailing design doctrine in neural QEC decoding and yields actionable principles for real-time deployment.
> > >
> > > We believe these insights are timely and valuable for both the physics and ML communities, and we respectfully encourage the reviewer to re-assess the contribution and potential impact in this light. We are happy to clarify any remaining points.
> > >
> > > Best regards,
> > >
> > > The authors

---

### Official Review · Reviewer_gg4N · 2026-03-19

**Soundness:** 3
**Presentation:** 3
**Significance:** 2
**Originality:** 3
**Overall Recommendation:** 4
**Confidence:** 3

**Summary:**

The paper argues that the inductive bias of neural decoders must be aligned with the spatiotemporal structure of quantum error correction (QEC) in order to scale well. By comparing multiple architectures—MLPs, CNNs (especially 3D CNNs), TCNs, Transformers, and GNNs—the authors show that models which explicitly exploit spatial locality and temporal correlations (e.g., 3D CNNs, TCNs, some Transformers) achieve good scalability and accuracy on QEC decoding, whereas architectures whose inductive biases are mismatched to the code geometry (MLPs, certain GNNs) fail to scale. They further study training efficiency and hardware deployment, demonstrating that 3D CNNs train faster than Transformers while reaching similar accuracy, and that a hardware-aware design with LUT-based INT4 quantization enables low-latency FPGA implementations by shifting from DSP-bound to logic-bound utilization, making neural decoders more practical for real-time QEC.

**Compliance With Llm Reviewing Policy:**

Affirmed.

**Key Questions For Authors:**

- How can we systematically design neural architectures whose inductive biases match irregular Tanner graphs or more complex QEC layouts (e.g., LDPC QEC codes, non-planar topologies)?

- Can we significantly reduce training cost and inference latency of Transformers (through sparse attention, structured weight sharing, low-rank, or better quantization) without losing decoding performance?

- What are the precise trade-offs between decoding accuracy, latency, and hardware resources when scaling to the sizes required for fault-tolerant quantum computing (e.g., thousands to millions of physical qubits)?

**Limitations:**

yes

**Strengths And Weaknesses:**

**Strengths**

- The authors evaluate multiple neural architectures (MLP, CNN, TCN, Transformer, GNN) on the same QEC task and consider not just accuracy but also training time and hardware latency, which is crucial for QEC in practice.

- The paper shows convincingly that aligning model inductive bias with QEC’s spatial and temporal structure is key to scalability.

- Demonstrates an implementation strategy that moves from DSP-bound to LUT/logic-bound usage on FPGAs, enabling lower latency.
Uses INT4 quantization while maintaining performance, which is important for embedded quantum control.

- Addresses concrete bottlenecks in moving QEC from theory to practice, especially under strict timing constraints.

**Weaknesses**

- The robustness of the decoders to different noise models or device drift is not deeply explored. It is unclear how sensitive the architectures are to mismatches between training and deployment conditions.

- Realistic QEC codes and fault-tolerant schemes can induce irregular or non-grid-like Tanner graphs. The paper offers limited guidance on how to build suitable inductive biases for such irregular structures.

- Transformers achieve good final accuracy but have high training cost and complexity. The work does not fully explore techniques (e.g., efficient attention, sparsity, low-rank) that could mitigate this.

---

> ### Author Rebuttal · Authors · 2026-03-31
>
> We sincerely thank the reviewer for the valuable comments. Below, we address the remaining concerns.
>
> # Response to W1 (Different noise models)
> We have conducted new experiments under SI1000 noise model (capturing realistic hardware asymmetries far beyond our original depolarizing setup), which introduces non-uniform error rates across different operations (gate=$p_{base}$, measure=$5p_{base}$, reset=$2p_{base}$, idle=$2p_{base}$) with $p_{base}$ being the base physical error rate. Following AlphaQubit1 and 2, here we set $p_{base}=0.004$.
>
> The results below consistently show that our neural decoders outperform the classical baselines.
>
> **SI1000 noise model (**$p_{base}$**=0.004), TCN decoder, 3 runs:**
>
> | Setting | LER_TCN (%) | LER_PyMatching (%) | Relative Improvement |
> | --- | --- | --- | --- |
> | d=5, small | 2.29 ± 0.05 | 2.84 ± 0.08 | 19.4% |
> | d=7, large | 1.37 ± 0.06 | 2.17 ± 0.05 | 36.9% |
>
> We would also highlight that a model trained in the critical regime (i.e., $p=0.005$; p_{base}=0.004$) can perform **zero-shot inference** with lower error rates without retraining, consistently outperforming PyMatching across all tested rates. These results reflect that neural decoders can exhibit **strong robustness to noise variations and device drift**.  Below, we show representative zero-shot results:
>
> **Uniform (trained at $p$=0.005), zero-shot across error rates:**
>
> | Target $p$ | d=5 LER_TCN (%) | d=5 LER_PyMatching (%) | d=7 LER_TCN (%) | d=7 LER_PyMatching (%) |
> | --- | --- | --- | --- | --- |
> | 0.005 | 1.03 ± 0.04 | 1.42 ± 0.01 | 0.56 ± 0.03 | 0.99 ± 0.03 |
> | 0.004 | 0.50 ± 0.03 | 0.76 ± 0.02 | 0.20 ± 0.02 | 0.43 ± 0.03 |
> | 0.003 | 0.21 ± 0.01 | 0.32 ± 0.00 | 0.06 ± 0.00 | 0.14 ± 0.00 |
> | 0.002 | 0.06 ± 0.00 | 0.10 ± 0.00 | 0.01 ± 0.00 | 0.03 ± 0.00 |
> | 0.001 | 0.01 ± 0.00 | 0.01 ± 0.00 | 0.00 ± 0.00 | 0.00 ± 0.00 |
>
> **SI1000 (trained at $p_{base}$=0.004), zero-shot across error rates:**
>
> | Target $p_{base}$ | d=5 LER_TCN (%) | d=5 LER_PyMatching (%) | d=7 LER_TCN (%) | d=7 LER_PyMatching (%) |
> | --- | --- | --- | --- | --- |
> | 0.004 | 2.29 ± 0.05 | 2.84 ± 0.08 | 1.37 ± 0.06 | 2.17 ± 0.05 |
> | 0.003 | 0.98 ± 0.02 | 1.28 ± 0.02 | 0.41 ± 0.00 | 0.72 ± 0.01 |
> | 0.002 | 0.28 ± 0.01 | 0.39 ± 0.01 | 0.08 ± 0.00 | 0.14 ± 0.00 |
> | 0.001 | 0.03 ± 0.00 | 0.05 ± 0.00 | 0.00 ± 0.00 | 0.01 ± 0.00 |
>
> Together with Table 1 and the supplementary results, we evaluate our decoders under diverse noise models, including uniform noise and the Sycamore DEM; the zero-shot tests provide preliminary evidence of robustness to noise-model mismatch and device drift.
>
> # Response to W2 & Q1 (Issues with irregular Tanner graphs)
> We deliberately focus on the surface code: while non-grid codes such as qLDPC are theoretically promising, they lack mature real-device DEMs and face substantial implementation hurdles. Thereby, we restrict our scope to the surface code, which is the leading candidate towards fault-tolerant quantum computing.
>
> # Response to W3 & Q2(**Efficient Transformer Design**)
> We identify a fundamental gap between conventional ML inference and the operational realities of QEC: decoding must finish within a microsecond-scale budget ($\sim 1 \mu s$), otherwise syndrome backlogs break fault tolerance. Since GPU/TPU inference typically operates at millisecond latency, real-time decoding necessitates FPGA/ASIC deployment. Accordingly, techniques such as sparse attention, low-rank approximation, and weight sharing, which are effective on GPU/TPU platforms, do not **directly translate to FPGA execution** and are unlikely to satisfy microsecond-scale constraints. This motivates our focus on hardware-aware and real-time neural decoders rather than accuracy-only optimization.
>
> In addition, Table 2 (main text) indicates that the bottleneck at d=9 is the overall MAC/parameter footprint, and Transformers already fall outside the feasibility boundary, whereas TCN remains viable. We therefore focus on INT4 quantization and pruning, which directly reduce FPGA footprint and are necessary for real-time deployment; attention-specific optimizations are unlikely to change this conclusion.
>
> # Response to Q3 (**Accuracy–latency–resource trade-offs**)
> The trade-off is mainly controlled by code distance d: accuracy improves with larger d, while latency and hardware cost increase because the syndrome volume and compute scale with $\Theta(d^2)$ detectors and time rounds. Our FPGA model links each decoder’s compute and memory demand to FPGA resource usage and the resulting microsecond-scale latency, allowing us to identify which code distances are feasible for real-time deployment (e.g., d $\le 9$ on a single card in our current design).
>
> Remark: Million-qubit scales mainly arise from circuit-level overheads (e.g., lattice surgery, routing, and multi-patch decoding), which are beyond the scope of our single-logical-qubit memory study. At current error rates, a surface-code logical qubit typically requires on the order of 1000 physical qubits.

---

### Decision · Program_Chairs · 2026-04-30

**Decision:**

Accept (regular)

**Comment:**

This paper systematically re-evaluates neural decoders for surface-code QEC under microsecond-scale FPGA latency constraints, comparing five architectures and showing that data scale dominates architectural complexity once appropriate inductive bias is present. The panel is unanimous at weak accept (4, 4, 4, 4). Reviewers recognize the practical value of the hardware-aware evaluation and the INT4 QAT finding (gg4N, Yd1n), and the rebuttal strengthened the empirical case with SI1000 experiments and HLS synthesis validation. The main recurring concern is whether the contribution is sufficiently novel beyond systematic benchmarking (uNpU, 2xJj), though both reviewers acknowledge the empirical soundness after rebuttal.